



Climate
of the Past

# Spiky fluctuations and scaling in high-resolution EPICA ice core dust fluxes

**Shaun Lovejoy[1] and Fabrice Lambert[2,3]**

[1]Physics Department, McGill University, 3600 University St., Montréal CE1, QC, H3A 2T8, Canada
[2]Department of Physical Geography, Pontificia Universidad Catolica de Chile, Vicuna Mackenna 4860, Santiago, Chile
[3]Millennium Nucleus Paleoclimate, Santiago, Chile TS1

**Correspondence:** Fabrice Lambert (lambert@uc.cl)

**Abstract.** Atmospheric variability as a function of scale has been divided in various dynamical regimes with alternating increasing and decreasing fluctuations: weather, macroweather, climate, macroclimate, and megaclimate. Although a vast amount of data are available at small scales, the larger picture is not well constrained due to the scarcity and low resolution of long paleoclimatic time series. Using statistical techniques originally developed for the study of turbulence, we analyse the fluctuations of a centimetric-resolution dust flux time series from the EPICA Dome C ice core in Antarctica that spans the past 800 000 years. The temporal resolution ranges from annual at the top of the core to 25 years at the bottom, enabling the detailed statistical analysis and comparison of eight glaciation cycles and the subdivision of each cycle into eight consecutive phases. The unique span and resolution of the dataset allows us to analyse the macroweather and climate scales in detail.

We find that the interglacial and glacial maximum phases of each cycle showed particularly large macroweather to climate transition scale $\tau_c$ (around 2 kyr), whereas mid-glacial phases feature centennial transition scales (average of 300 years). This suggests that interglacials and glacial maxima are exceptionally stable when compared with the rest of a glacial cycle. The Holocene (with $\tau_c \approx 7.9$ kyr) had a particularly large $\tau_c$, but it was not an outlier when compared with the phases 1 and 2 of other cycles.

We hypothesize that dust variability at larger (climate) scales appears to be predominantly driven by slow changes in glaciers and vegetation cover, whereas at small (macroweather) scales atmospheric processes and changes in the hydrological cycles are the main drivers.

For each phase, we quantified the drift, intermittency, amplitude, and extremeness of the variability. Phases close to the interglacials (1, 2, 8) show low drift, moderate intermittency, and strong extremes, while the "glacial" middle phases 3–7 display strong drift, weak intermittency, and weaker extremes. In other words, our results suggest that glacial maxima, interglacials, and glacial inceptions were characterized by relatively stable atmospheric conditions but punctuated by frequent and severe droughts, whereas the mid-glacial climate was inherently more unstable.

## 1 Introduction

Over the late Pleistocene, surface temperature variability is strongly modulated by insolation, both at orbital (Jouzel et al., 2007) and daily timescales. In between these two scales, temperature variability has been shown to scale according to power law relationships, thus evidencing a continuum of variability at all frequencies (Huybers and Curry, 2006). However, although a vast amount of high-resolution data exist for modern conditions, our knowledge of climatic variability at glacial–interglacial timescales is usually limited by the lower resolution of paleoclimatic archive records, thus restricting high-frequency analyses during older time sections. Previous analyses using marine and terrestrial temperature proxies from both hemispheres suggest a generally stormier and more variable atmosphere during glacial times than during interglacials (Ditlevsen et al., 1996; Rehfeld et al., 2018).

One of the difficulties in characterizing climate variability is that ice core paleotemperature reconstructions rapidly lose

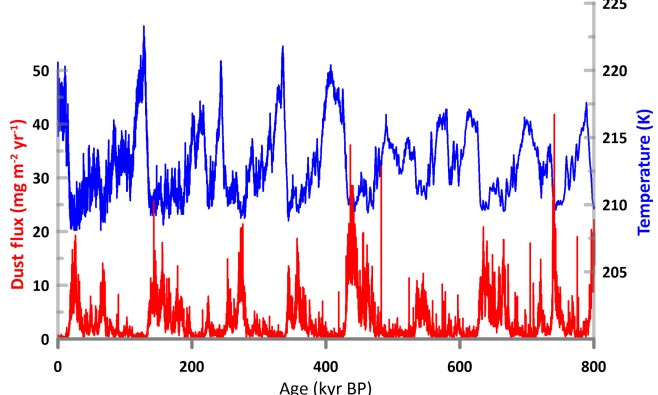

**Figure 1.** TS2 Temperature (blue) and dust flux (red) from the EPICA Dome C ice core (Jouzel et al., 2007; Lambert et al., 2012a). The dust flux time series has 32 000 regularly spaced points (25-year resolution); the temperature series has 5752 points. The temperature data are irregularly spaced and lose resolution as we go back into the past (number of temperature data points in successive ice ages: 3022, 1117, 521, 267, 199, 331, 134, 146). In both cases we can make out the glacial cycles, but they are at best only quasi-periodic.

their resolutions as we move to the bottom of the ice column. Figure 1 shows this visually for the EPICA Dome C Antarctic ice core temperature proxy (5787 measurements in all); the curve becomes noticeably smoother as we move back in time. In terms of data points, the most recent 100 kyr period has more than 3000 points ($\approx$ 30-year resolution), whereas the most ancient 100 kyr period has only 137 ($\approx$ 730-year resolution). This implies that while the most recent glacial–interglacial cycle can be perceived with reasonable detail, it is hard to compare it quantitatively to previous cycles or to deduce any general cycle characteristics.

Fluctuation analysis (Lovejoy, 2017; Lovejoy and Schertzer, 2013; Nilsen et al., 2016) gives a relatively simple picture of atmospheric temperature variability (Fig. 2). The figure shows a series of regimes each with variability alternately increasing and decreasing with scale. From left to right we see weather-scale variability, in which fluctuations tend to persist, building up with scale (they are unstable) and increasing up to the lifetime of planetary structures (about 10 d). This is followed by a macroweather regime with fluctuations tending to cancel each other out, decreasing with scale and displaying stable behaviour. In the last century, anthropogenically forced temperature changes (mostly from greenhouse gases) dominate the natural (internal macroweather) variability at scales longer than about 10–20 years. The figure shows that in pre-industrial periods, the lower-frequency climate regime starts somewhere between 100 and 1000 years (the macroweather–climate transition scale $\tau_c$), indicating that different long-frequency processes become dominant. The macroweather–climate transition scale marks a change of

regime where the dominant high-frequency processes associated with weather processes (and reproduced by GCMs in control runs) give way to a different regime, where the variability is dominated by either the responses to external forcings or to slow internal sources of variability that were too weak to be important at higher frequencies. Further to the right of Fig. 2, we can see the broad peak associated with the glacial cycles at about 50 kyr (half the 100 kyr period), and then at very low frequencies, the megaclimate regime again shows increasing variability with scale. In between the climate and megaclimate regimes, the fluctuations decrease with scale over a relatively short range from about 100 to 500 kyr. However, the temperature fluctuations shown in Fig. 2 display average behaviour, which can potentially hide large variations from epoch to epoch. In this paper, we use a uniquely long and high-resolution paleo-dataset to analyse the macroweather and climate scales in detail.

We focus on the EPICA Dome C dust flux record, which has a 55 times higher resolution than the deuterium record, including high resolution over even the oldest cycle (Lambert et al., 2012a, Fig. 1). Antarctic dust fluxes are well correlated with temperature at orbital frequencies (Lambert et al., 2008; Ridgwell, 2003). But the fluxes are also affected by climatic conditions at the source and during transport (Lambert et al., 2008; Maher et al., 2010). The dust data used here can therefore be thought of as a more "holistic" climatic parameter that includes not only temperature changes but describes atmospheric variability as a whole (including wind strength and patterns and the hydrological cycle).

## 2   Method

In order to proceed to a further quantitative analysis of the types of statistical variability and of the macroweather–climate transition scale, we need to make some definitions. A commonly used way of quantifying fluctuations is the Fourier analysis. It quantifies the contribution of each frequency range to the total variance of the process. However, the interpretation of the spectrum is neither intuitive nor straightforward (Sect. 2.3). The highly non-Gaussian spikiness for both dust flux and its logarithm (e.g. Fig. 3b, c), implies strong – but stochastic – Fourier space spikes. Indeed, Lovejoy (2018) found that the probability distributions of spectral amplitudes can themselves be power laws. This has important implications for interpreting spectra, especially those estimated from single series ("periodograms"): if the spectral amplitudes are highly non-Gaussian, then we will typically see strong spectral spikes whose origin is purely random. This makes it very tempting to attribute quasi-oscillatory processes to what are in fact random spectral peaks. It therefore makes sense to consider the real (rather than Fourier) space variability (fluctuations). The problem here is that the spectrum is a second-order statistical moment (the spectrum is the Fourier transform of the autocorre-

Clim. Past, 15, 1–19, 2019

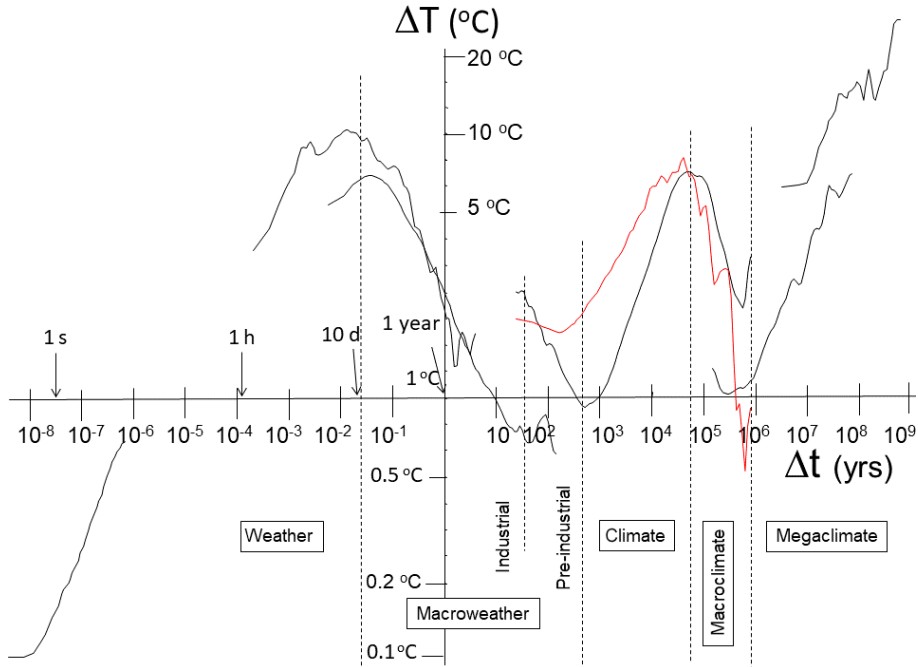

**Figure 2.** A composite showing root mean square (rms) Haar fluctuations ($\Delta T$ in units of °C) black and rms dust fluctuations analysed in this paper (red, in units of milligrams per square metre per year; Lambert et al., 2012a). From left to right: thermistor temperatures at 0.0167 s resolution (Lovejoy, 2018), hourly temperatures from Landers, Wyoming (Lovejoy, 2015), daily temperatures from 75° N (Lovejoy, 2015), EPICA Dome C temperatures (Jouzel et al., 2007), and two marine benthic stacks (Veizer et al., 1999; Zachos et al., 2001). The macroweather–climate transition is not in phase between the different records because the left ones (industrial side) are influenced by anthropogenic climate change, while the right data are pre-industrial natural variability. As elsewhere in this paper, the fluctuations were multiplied by the canonical calibration constant of 2 so that when the slopes are positive, the fluctuations are close to difference fluctuations. The various scaling regimes are indicated at the bottom. Adapted from Lovejoy (2017).

lation function). While second-order moments are sufficient for characterizing the variability of Gaussian processes, in the more general and usual case – especially with the highly variable dust fluxes – we need to quantify statistics of higher orders, in particular, the higher-order statistics that characterize the extremes. Here, we will use two simple concepts to describe variability and intermittency (or spikiness) of the data.

The theoretical framework that we use in this paper is that of scaling, multifractals, and the outcome of decades of research attempting to understand turbulent intermittency. Intermittent, spiky transitions – characterized by different scaling exponents for different statistical moments – turn out to be the generic consequence of turbulent cascade processes. Although the cascades are multiplicative, the extreme probabilities generally turn out to be power laws (Mandelbrot, 1974; Schertzer and Lovejoy, 1987), not log-normals (as was originally proposed by Kolmogorov, 1962). The analyses are based on scaling regimes and their statistical characteristics. Because scaling is a symmetry (in this case invariance of exponents under dilations in time), in a dynamical regime in which two different components – such as temperature and dust – are strongly coupled parts of the system, each may

have different scaling properties but both should respect the scale symmetry including the transition scale at which the symmetry breaks down. Therefore, the broad conclusions of our dust flux analyses – scaling regimes and their break points and stability or instability – are expected to be valid for the more usual climate parameters including the temperature. Although it is beyond our present scope, we will explore the scale-by-scale relationship between EPICA dust fluxes and temperatures in a future publication.

## 2.1 Haar fluctuations

The basic tool we use to characterize variability in real space is the Haar fluctuation, which is simply the absolute difference of the mean over the first and second halves of an interval:

$$\Delta F(\Delta t) = \frac{2}{\Delta t} \int_{t-\Delta t/2}^{t} F(t')\,\mathrm{d}t' - \frac{2}{\Delta t} \int_{t-\Delta t}^{t-\Delta t/2} F(t')\,\mathrm{d}t'. \quad (1)$$

We can characterize the fluctuations by their statistics. For example, by analysing the whole dataset using intervals of various lengths, we can thus define the variability as

a function of scale (i.e. interval length). If over a range of timescales $\Delta t$, there is no characteristic time, then this relationship is a power law, and the mean absolute fluctuation varies as

$$\langle |\Delta F(\Delta t)| \rangle \propto \Delta t^H, \tag{2}$$

where "$\langle \rangle$" indicates ensemble average – here an average over all the available disjointed intervals. A positive $H$ implies that the average fluctuations increase with scale. This situation corresponds to unstable behaviour identified with the climate regime. In contrast, when $H$ is negative, variability converges towards a mean state with increasing scale. This is the situation found in the stable macroweather regime. Haar fluctuations are useful for the exponent range $-1 < H < 1$, which is valid for the dust series, and indeed for almost all geodata analysed to date.

More generally, we can consider other statistical moments of the fluctuations, the "generalized structure functions", $S_q(\Delta t)$:

$$S_q(\Delta t) = \langle |\Delta F(\Delta t)|^q \rangle \propto \Delta t^{\xi(q)}, \tag{3}$$

If the fluctuations are from a Gaussian process, then their exponent function is linear: $\xi(q) = qH$. More generally however, $\xi(q)$ is concave and it is important to characterize this, since the non-linearity in $\xi(q)$ is due to intermittency, i.e. sudden, spiky transitions (for more details on Haar fluctuations and intermittency, we refer to Lovejoy and Schertzer, 2012). We therefore decompose $\xi(q)$ into a linear and a non-linear (convex) part $K(q)$, with $K(1) = 0$:

$$\xi(q) = qH - K(q), \tag{4}$$

so that $K(q) = 0$ for quasi-Gaussian processes. Since the spectrum is a second-order moment, the spectrum of a scaling process at frequency $\omega$ is a power law:

$$E(\omega) \approx \omega^{-\beta}, \tag{5}$$

where the spectral exponent $\beta = 1 + \xi(2) = 1 + 2H - K(2)$; $K(2)$ is therefore sometimes termed the "intermittency correction".

## 2.2 Intermittency

A simple way to quantify the intermittency is thus to compare the mean and root mean square (rms) Haar fluctuations:

$$S_1(\Delta t) = \left\langle |(\Delta F(\Delta t))| \right\rangle \propto \Delta t^{\xi(1)} = \Delta t^H, \tag{6}$$

$$S_2(\Delta t)^{1/2} = \left\langle (\Delta F(\Delta t))^2 \right\rangle^{1/2} \propto \Delta t^{\xi(2)/2} = \Delta t^{H-K(2)/2}, \tag{7}$$

with the ratio

$$S_1(\Delta t)/S_2(\Delta t)^{1/2}$$
$$= \left\langle |\Delta F(\Delta t)| \right\rangle \Big/ \left\langle (\Delta F(\Delta t))^2 \right\rangle^{1/2} \propto \Delta t^{K(2)/2}, \tag{8}$$

where we estimate $S(\Delta t)$ using all available disjointed intervals of size $\Delta t$. These expressions are valid in a scaling regime. Since the number of disjointed intervals decreases as $\Delta t$ increases, so does the sample size; hence, the statistics are less reliable at large $\Delta t$.

For theoretical reasons (Lovejoy and Schertzer, 2013; Schertzer and Lovejoy, 1987), it turns out that the intermittency near the mean ($q = 1$) is best quantified by the parameter $C_1 = K'(1)$. Since $K(1) = 0$ is a basic property, it turns out that for log-normal multifractals (approximately relevant here), the ratio exponent $K(2)/2 \approx C_1$.

While the mean-to-rms ratio is an intuitive statistic, it does not give a direct estimate of $C_1$: a more accurate estimate of $C_1$ uses the intermittency function $G(\Delta t)$:

$$G(\Delta t) = \lim_{\Delta q \to 0} \langle \Delta F \rangle \left[ \frac{\langle \Delta F^{1-\Delta q} \rangle}{\langle \Delta F^{1+\Delta q} \rangle} \right]^{1/(2\Delta q)}$$
$$\propto \Delta t^{\xi(1)-\xi'(1)} = \Delta t^{C_1} \tag{9}$$

(this is exact in the limit $\Delta q \to 0$), whose exponent is $C_1$. The intermittency exponent $C_1$ quantifies the *rate* at which the clustering near the mean builds up as a function of the range of scales over which the dynamical processes act; it only partially quantifies the spikiness. For this, we need other exponents, in particular the exponent $q_D$ that characterizes the tails of the probability distributions. This is because scaling in space and/or time generically gives rise to power law probability distributions (Mandelbrot, 1974; Schertzer and Lovejoy, 1987). Specifically, the probability ($Pr$) of a random dust flux fluctuation $\Delta F$ exceeding a fixed threshold $s$ is

$$Pr(\Delta F > s) \approx s^{-q_D}; \quad s \gg 1, \tag{10}$$

where the exponent $q_D$ characterizes the extremes; for example, $q_D \approx 5$ has been estimated for wind or temperature (Lovejoy and Schertzer, 1986) and for paleotemperatures (Lovejoy and Schertzer, 2013), whereas $q_D = 3$ for precipitation (Lovejoy et al., 2012). A qualitative classification of probability distributions describes classical exponentially tailed distributions (such as the Gaussian) as "thin-tailed", log-normal (and log-Lévy) distributions as "long-tailed", and power law distributions as "fat-tailed". Whereas thin and long-tailed distributions have convergence of all statistical moments, power distributions only have finite moments for orders $q < q_D$.

## 2.3 How fluctuations help interpret spectra

Although spectra may be familiar, their physical interpretations are nontrivial, a fact that was underscored in Lovejoy (2015). In a scaling regime – a good approximation to the macroweather and climate regimes discussed here – the spectrum is a power law form (Eq. 5) where the spectral exponent $\beta$ characterizes the spectral *density*. Although $\beta$ tells us how

quickly the variance changes per frequency *interval*, its physical significance is neither intuitive nor obvious. Integrating the spectrum over a frequency range is already easier to understand; it is the total variance of the process contributed by the range. Therefore, we already see that $\beta - 1$ (the exponent of the integrated spectrum) is more directly relevant than $\beta$. But even to understand this, we need to consider whether over a range of frequencies, the process is dominated by high or low frequencies. For this, we can compare the total variance contributed by neighbouring octaves. For a power law spectrum, the variance ratio of one octave to its neighbouring higher-frequency octave is $2^{1-\beta}$. From this, we see that $\beta > 1$ yields a ratio $2^{1-\beta} < 1$ implying low-frequency dominance, whereas when $\beta < 1$, we have $2^{1-\beta} > 1$ and high-frequency dominance.

But what does low-frequency or high-frequency "dominance" mean physically? For this, it is easier to consider the situation in real space using fluctuations; the simplest relevant fluctuations are the Haar fluctuations $\Delta F$ discussed in Sect. 2.2 that vary with time interval $\Delta t$ as $\Delta F \approx \Delta t^H$. We saw that the exponents in real and spectral space were simply related by $\beta = 1 + 2H - K(2)$, where $K(2) > 0$ due to the spikiness (intermittency). This formula leads to two important conclusions. First, if we ignore intermittency (putting $C_1 = 0$, hence $K(2) = 0$) and assume that the mean fluctuations scale with the same exponent as the rms fluctuations, then $H = (\beta - 1)/2$, showing again that it is the sign of $\beta - 1$ that is fundamental: $\beta > 1$ implies $H > 0$; hence, fluctuations grow with scale and the process "drifts" or "wanders"; it is unstable. Conversely $\beta < 1$ implies $H < 0$; hence, fluctuations decrease with scale and the process "cancels" and "converges"; it is "stable". The second conclusion is that if intermittency is strong (here we typically have $C_1 \approx 0.1$, $K(2) \approx 0.2$), then the relationship between the second- and first-order statistical moments is a little more complex so that for example, with these values and a $\beta \approx 0.9$, we would have high frequencies dominating the variance ($\beta < 1$) but low frequencies dominating the mean ($H > 0$).

## 2.4 Dust flux data

The dust flux data used in this study are based on a linear combination of insoluble particles, calcium, and non-sea-salt calcium concentrations (Lambert et al., 2012a). Because missing-data gaps in the three original datasets were linearly interpolated prior to the PCA, high-frequency variability can sometimes be underestimated in short sections that feature a gap in one of the three original datasets. This occurs in about 25 % of all dust flux data points, although half of those are concentrated in the first 760 m of the core (0–43 kyr BP), when an older, less reliable dust-measuring device was used. Below 760 m these occurrences are evenly distributed and do not affect our analysis. Due to the sometimes slightly underestimated variability, the analysis shown here is a conservative estimate (Lambert et al., 2012a).

Unlike water isotopes that diffuse and lose their temporal resolution in the bottom section of an ice core at high pressures and densities, the relatively large dust particles diffuse much less and have been used to estimate the dust flux over every centimetre of the 3.2 km long EPICA core (298 203 valid data points; Lambert et al., 2012a). The temporal resolution of this series varies from 0.81 to 11.1 years (the averages over the most recent and the most ancient 100 kyr respectively). The worst temporal resolution of 25 yr cm$^{-1}$ occurs around 3050 m depth, with the result that at that resolution, there are virtually no missing data points in the whole record (Fig. 1). We note, however, that the dust flux used here is a construct of concentrations at 1 cm resolution and accumulation rates at 55 cm resolution that were linearly interpolated to match the dust concentration resolution.

## 3 Results

### 3.1 Looking at the data

Polar dust flux records cannot be assigned to one particular atmospheric variable. At any given moment, the amount of dust deposited in East Antarctica will depend on the size and vegetation cover of the source region (mostly Patagonia for East Antarctic dust; Delmonte et al., 2008), on the amount of dust available in the source region (can depend on the presence of glaciers), on the strength of the prevailing winds between South America and Antarctica, and on the strength of the hydrological cycle (more precipitation will wash out more dust from the atmosphere; Lambert et al., 2008). Over large scales it is thought that temperature-driven moisture condensation may be the major process driving low-frequency variability (Markle et al., 2018), although that may not be true everywhere (Schüpbach et al., 2018). High- and low-frequency variability in the dust flux record is likely driven by different processes. For example, dust source conditions related to glaciers and vegetation cover may not have influenced high-frequency variability due to their relatively slow rate of change. On the other hand, volcanic eruption or extreme events related to the hydrological cycle may produce high-frequency signals in the record. A single dust peak within a low background may therefore reflect a short-term atmospheric disturbance like an eruption or drought over South America or low precipitation over the Southern Ocean. The analysis presented here focuses heavily on the occurrence of dust fluctuations, the physical interpretation of which will depend on the scale of the phenomenon.

Figure 3a shows a succession of 10 factor-of-2 "blow downs" (upper left to lower right at 11 different resolutions). In order to avoid smoothing, the data were "zoomed" in depth rather than time, but the point is clear: the signal is very roughly scale invariant, at no stage is there any sign of obvious smoothing, and the quasi-periodic 100 kyr oscillations are the only obvious timescale (we quantify this below). In

Please note the remarks at the end of the manuscript.

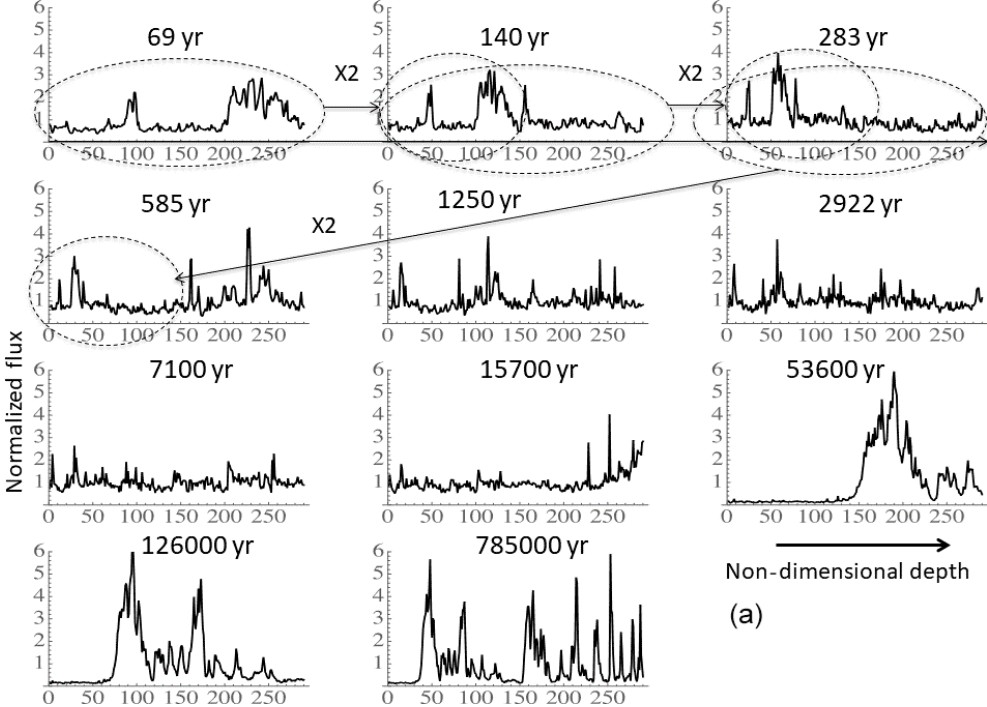

**Figure 3.**

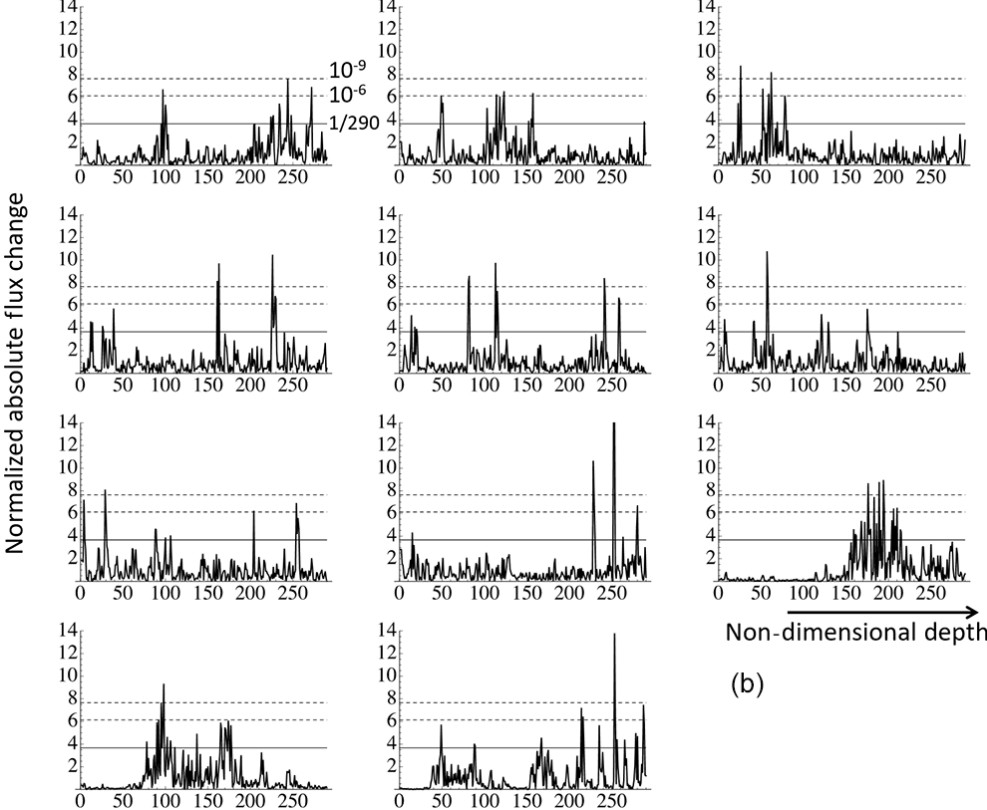

**Figure 3.**

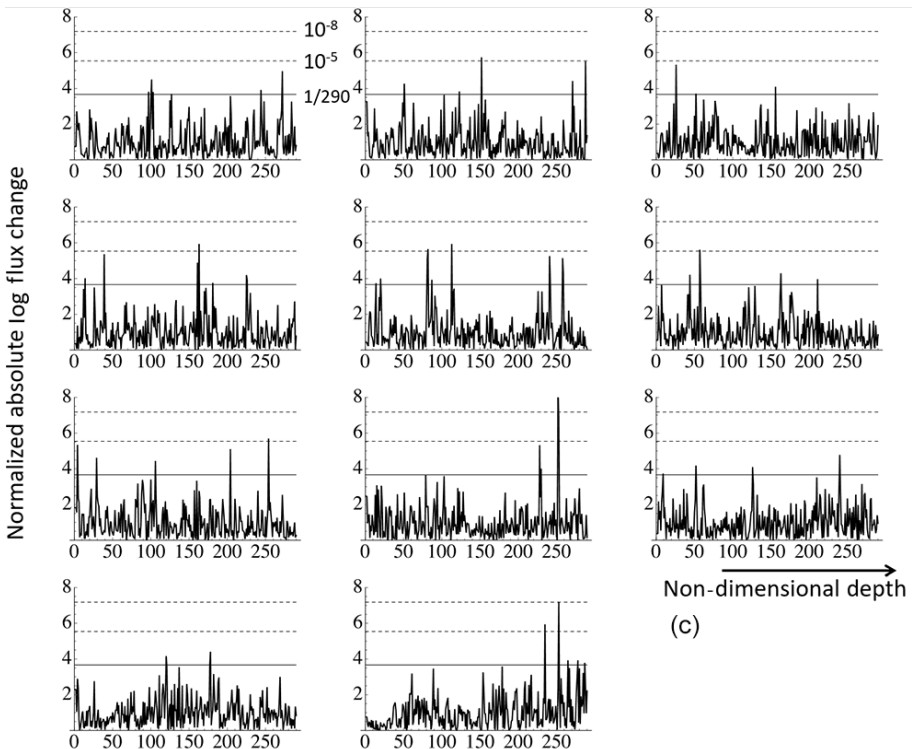

**Figure 3. (a)** Zooming out of the Holocene dust fluxes by octaves, by doubling the depth resolution from 1 cm (upper left) to 11 m (lower right) resolution. Starting at the left and moving to the right and from top to bottom (see the ellipses on the first three in the sequence), we zoom out by factors of 2 in depth maintaining exactly 290 data points (effectively non-dimensionalizing the depth; the small number of missing data points were not interpolated so that the final resolution is not exactly $2^{10}$ cm $= 10.24$ m). The temporal resolution is not exactly doubled due to the squashing of the ice column; the total duration (in years) of each section is indicated in each plot; the average temporal resolution of plots is 0.24, 0.48, 0.98, 2.02, 4.32, 10.1 24.5, 54.1, 184, 434, and 2710 years. In order to fit all the curves on the same vertical scale, the dust fluxes were normalized by their mean over each segment. The means (in milligrams per square metre per year) are 0.44, 0.38, 0.30, 0.36, 0.35, 0.33, 0.34, 0.39, 2.48, 2.18, and 2.41, i.e. the first eight plots have nearly the same vertical scales, whereas the last three are about 6 times larger in range. This means that all the plots except the last three are at nearly constant normalization. **(b)** Same as **(a)** but for the absolute changes between neighbouring values in dust flux normalized by the corresponding mean over the segment (290 points). The horizontal lines indicate the Gaussian probability levels for $p = 1/290$ (representing the mean extreme for a 290-point segment; full line) as well as $p = 10^{-6}$ (lower dashed line) and $p = 10^{-9}$ (upper dashed line). **(c)** Same as **(a)** but for the absolute changes between neighbouring values in the logarithms of dust flux normalized by the corresponding mean over the segment (290 points). The horizontal lines indicate the Gaussian probability levels for $p = 1/290$ (representing the mean extreme for a 290-point segment; full line) as well as $p = 10^{-5}$ (lower dashed line) and $p = 10^{-8}$ (upper dashed line, not the same as in **b**).

comparison with more common paleoclimate signals, such as temperature proxies – which are apparently smoother but with spiky transitions – the dust flux itself is already quite spiky. However, it also displays spiky transitions. In Fig. 3b we show the absolute change in dust flux, and one can visually see the strong spikiness associated with strongly non-Gaussian variability: the intermittency. At each resolution, the solid line indicates the maximum spike expected if the process was Gaussian, and the upper dashed lines show the expected level for a (Gaussian) spike with probability $10^{-6}$. Again, without sophisticated analysis, we can see that the spikes are wildly non-Gaussian, frequently exceeding the $10^{-6}$ level even though each segment has only 290 points, with the spikiness being nearly independent of resolution.

Taking the logarithms of the dust flux is common practice since it reduces the extremes and makes the signal closer to the temperature and other more familiar atmospheric parameters. We therefore show the corresponding spike plot for the log-transformed data (Fig. 3c). Although the extreme spikes are indeed less extreme (see also Fig. 6a, b), we see that the transformation has not qualitatively changed the situation, with spikes still regularly exceeding (log-) Gaussian probability levels of $10^{-5}$ and occasionally $10^{-8}$.

## 3.2 Spectra

Figure 4 shows various spectral analyses (for the corresponding fluctuation analyses, see Fig. 5). There is a clear periodicity at about $(100\,\text{kyr})^{-1}$. In the double power law fit

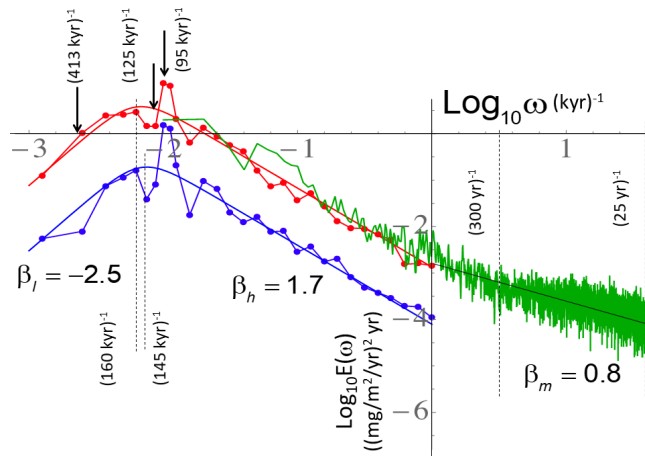

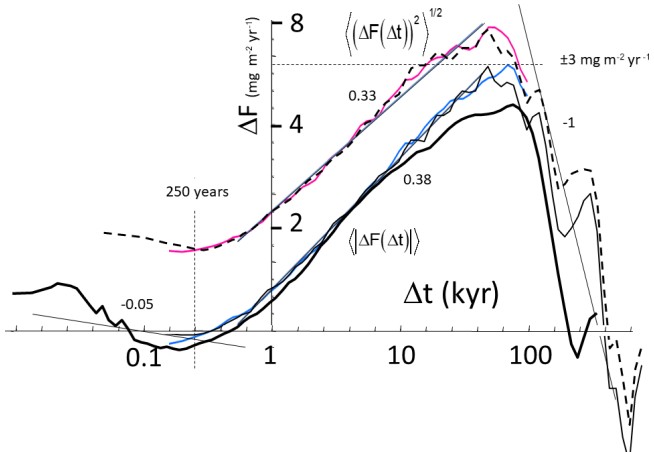

**Figure 4.** Log–log plot of the Fourier spectrum of the $(25\ \text{year})^{-1}$ resolution dust concentration in frequency units of $\text{kyr}^{-1}$ (red) and the same but of the logarithms of the flux (blue). Also shown is the average spectrum of the 5-year resolution data over the last 400 kyr (green). For the latter, the periodograms of each the four most recent 100 kyr cycles were averaged, but the full spectral resolution $(5\ \text{years})^{-1}$ was retained. The beta parameters are the exponents of the theoretical spectrum (see main text, the negative of the logarithmic slope) for the macroclimate $(-2.5)$, climate $(1.7)$, and macroweather $(0.8)$ regimes. The spectra were analysed using FFT CE3 with standard Hanning windows.

(line plot), the transition frequencies are a little lower: $\omega_0 = (16)\text{kyr})^{-1}$ (flux) and $\omega_c = (145\ \text{kyr})^{-1}$ (log flux), although a Gaussian fit near the max gives a spike at $(94 \pm 9\ \text{kyr})^{-1}$. Note that it is actually a little bit "wide" (two peaks); hence, it is not perfectly periodic, and the amplitude is only about a factor 4 above the background. In comparison, the amplitude of the annual temperature frequency peak is several thousand times above the background (depending on the location) and is narrower (not shown).

Since this is a log–log plot, power laws appear as straight lines. We show in the figure the fits to the bi-scaling function

$$E(\omega) = \frac{a}{(\omega/\omega_0)^{\beta_h} + (\omega/\omega_0)^{\beta_l}} \tag{11}$$

that smoothly transitions between a spectrum with $E(\omega) \approx \omega^{-\beta_h}$ at $\omega > \omega_0$ and $E(\omega) \approx \omega^{-\beta_l}$ at $\omega < \omega_0$. The figure shows the regressions with $\beta_l = -2.5$, $\beta_h = 1.7$, and $a = 7.5\ (\text{mg m}^{-2}\,\text{yr}^{-1})^2\,\text{yr}$ TS5, $\omega_0 \approx (145\ \text{kyr})^{-1}$ for the fluxes, and $a = 0.375\ \text{years}^{-1}$, $\omega_0 \approx (160\ \text{kyr})^{-1}$ for the logarithms of fluxes. According to the figure, the high-frequency climate regime scaling continues to about $(300\ \text{years})^{-1}$ before flattening to a very high-frequency scaling ($\beta_m \approx 0.8$) "macroweather" regime (Lovejoy and Schertzer, 2013). The scaling exponents $\beta_h = 1.7$ and $\beta_m = 0.8$ corresponding to the climate and macroweather regime respectively may be compared with the values 2.1 and 0.4 for the EPICA paleotemperatures discussed in a future publication (compare, however, the red and black curves in Fig. 2). These results

**Figure 5.** The Haar fluctuation analysis of the entire 800 kyr dust flux dataset (thin lines). The dashed black and solid pink lines (top pair) represent rms fluctuations for dimensional and non-dimensional time respectively. The solid black and blue curves are the same but for the mean absolute ($q = 1$) fluctuations. The curves with non-dimensional time lags have nominal (average) resolutions of 25 years and the fluctuation statistics are averaged over the eight cycles. The thick black line shows the Haar fluctuations for the most recent 400 kyr at a 5-year resolution. Note that the peak in the curves occurs as expected at $\Delta t \approx 50\ \text{kyr}$, i.e. at about a half cycle; the horizontal dashed line shows that at this scale – corresponding to the largest difference in phases – the change in the mean absolute dust flux is about $\pm 3\ \text{mg m}^{-2}\,\text{yr}^{-1}$. Also shown (dashed vertical line) is the (average) timescale $\tau_c \approx 250\ \text{years}$ at which the transition from macroweather to climate occurs. Several reference lines (with the slopes or exponents indicated) are shown showing approximate scaling behaviours.

show that temperature and dust variability are of the same statistical type so that it is likely that the dust signal is a real climate signal – yet the significant differences in their exponents shows that it has a different information content.

The variability shown in Fig. 4 can be interpreted broadly or in detail. A clear feature is the spectral maximum at around $(100\ \text{kyr})^{-1}$. The broad bispectral scaling model (Eq. 11) of the peak already accounts for 96 % of the spectral energy (variance), leaving only 4 % for the (extra) contribution from the (near-) $(100\ \text{kyr})^{-1}$ orbital frequency (using the logarithm of the flux changes little CE4). Alternatively, with a narrow Gaussian-shaped spectral spike model, the spike is localized at $(94 \pm 9\ \text{kyr})^{-1}$ and contributes a total of 31 % of the total variance. However, not all of this is above what we would expect from a scaling background; the exact amount depends on how the background is defined. For example, over the range from the 6th- to the 11th-highest frequencies in this discrete spectrum (from $(133\ \text{kyr})^{-1}$ to $(72\ \text{kyr})^{-1}$), in comparison to the background over this range, there is an enhancement of about 80 % due to the strong peaks (the enhancement is about 100 % for the 7th to the 12th frequencies). This means that, although the $(94 \pm 9\ \text{kyr})^{-1}$ peak rep-

resents 31 % of the total variability over the range from $(800\,\text{kyr})^{-1}$ to $(25\,\text{years})^{-1}$, it is only about 15 % above the "background" (note that only 5 % of the total variance is between $(25\,\text{years})^{-1}$ and $(1\,\text{kyr})^{-1}$). We did not do the corresponding analysis for the $(41\,\text{kyr})^{-1}$ obliquity frequency since Fig. 4 shows visually that it is barely discernable above the background.

The overall conclusion is that the background represents between 85 % and 96 % of the total variance.

## 3.3 Haar fluctuation analysis

Figure 5 shows the Haar fluctuations comparing their statistics for both dimensional and non-dimensional cycles as well as for the mean and rms fluctuations (bottom and top set of curves respectively). To start, let us consider the direct interpretations of the fluctuations in terms of the variability of the dust flux. Recall that when the fluctuations increase with scale, they represent typical differences, whereas when then decrease with scale, they represent typical anomalies (deviations from long-term mean values). For example, typical variations over a glacial–interglacial cycle (half cycle $\approx 50\,\text{kyr}$) are about $\pm 3\,\text{mg m}^{-2}\,\text{yr}^{-1}$ (i.e. a range of $6\,\text{mg m}^{-2}\,\text{yr}^{-1}$, the dashed horizontal line), whereas typical variations at the 250-year minimum are $\approx \pm 0.5\,\text{mg m}^{-2}\,\text{yr}^{-1}$.

The macroweather, climate, and macroclimate regimes noted in Fig. 4 are also clearly visible. In Fig. 5, we can clearly see the short regime with $H < 0$ (up to about 250 years), a scaling regime with $H > 0$ (up to glacial–interglacial periods $\approx 50\,\text{kyr}$), and finally a long-time (possibly scaling) decrease in variability. The spectral and real-space statistics are linked via the relation $\beta = 1 + \xi(2)$ (see Eqs. 4, 5). Starting with the high-frequency macroweather regime, the exponents $H = -0.05$, $K(2) \approx 0.10$ correspond to $\beta = 0.8$ (Fig. 4) and the real-space macroweather–climate transition scale ($\tau_c \approx 250\,\text{years}$) is close to the spectral transition scale ($1/\omega_c \approx 300\,\text{years}$, Fig. 4). In the middle (climate) regime, the top (rms) curves (slope 0.33) implies $\xi(2) = 0.66$, $\beta = 1.66$, which is close to the corresponding exponent in Fig. 4 ($\beta_h = 1.7$). Finally, at the longest (macroclimate) scales, the low-frequency part of the spectrum in Fig. 4 ($\beta_l = -2.5$) implies that the fluctuation exponent $H \approx (\beta_l - 1)/2 = -1.75$. However, this is less than the minimum detectable by Haar analysis ($H = -1$); therefore, we expect the far-right slope to equal $-1$ (as shown by a reference line). To correctly estimate this steep slope, one must use other definitions of fluctuations. We could also note that the climate–macroclimate transition timescale is broad and a little shorter than the value spectral value $1/\omega_c$ estimated in Fig. 4.

Beyond confirming the results of the spectral analysis and allowing for direct interpretations of the fluctuation values in terms of typical fluxes, Haar analysis also quantifies the intermittency from the convergence of the rms and mean statis-

tics at larger and larger timescales (see the clear difference in slopes shown in the climate regime: 0.38 versus 0.33). This underlines the limitation of spectral analysis discussed earlier: the fact that it is a second-order statistic that is only a partial characterization of the variability. Finally, the figure also shows that regardless of whether the cycles are defined in dimensional or in non-dimensional time, statistical characterizations (including the exponents) are virtually unaffected.

Figure 6a shows the fluctuation probabilities of the entire 800 kyr series at a 25-year resolution (here the fluctuations are simply taken as absolute differences at a 25-year resolution). We see that the large fluctuations (the tail) part of the distribution is indeed quite linear on a log–log plot, with exponents $q_D \approx 2.75$ and 2.98 in time and depth respectively (both from fits to the extreme 0.1 % of the distributions). To get an idea of how extreme these distributions are, consider the depth distribution with $q_D = 2.98$. With this exponent, dust flux fluctuations 10 times larger than typical fluctuations occur only $10^{2.98} \approx 1000$ times less frequently. In comparison, for a Gaussian, they would be $\approx 10^{23}$ times less likely; they would never be observed.

While the dust fluxes are always positive and so cannot be Gaussians, the increments analysed here could easily be approximately so. Nevertheless, a common way of trying to tame the spikes is by making a log transformation of fluxes. Figure 4 already showed that this did not alter the spectrum very much; here it similarly has only a marginal effect. For example, Fig. 6b shows that the extreme tails on the log dust flux distribution has $q_D = 3.60$ in time (25 years) and 4.59 in depth (at 1 cm resolution). The log-transformed variable still displays huge extremes with the extreme log flux corresponding to a log-Gaussian probability of $10^{-30}$ and $10^{-50}$ (time and depth respectively). Whether or not taking logarithms yields a more climate-relevant parameter, it does not significantly change the problem of intermittency or of the extremes.

We must mention the problem of estimating the uncertainties in the exponents. In the familiar case, we test a deterministic model and then uncertainty estimates are based on a stochastic model of the errors which are often assumed to be independent Gaussian random variables. In our case, the basic model is a stochastic one, and therefore one needs a stochastic model of the underlying process from which one can draw random time series. While our paper aims to provide a basis for the formulation of such a model, it is beyond our present scope. In order to obtain robust conclusions, we instead rely primarily on cycle-to-cycle comparisons, two different definitions of time (dimensional and non-dimensional) as well as a diversity of analysis techniques (spectral, fluctuation analysis, probability distributions). We should also mention that the use of fluxes (product of 1 cm concentrations and 55 cm accumulation rate) introduces an additional source of uncertainty due to the different time ranges contained in these sections at various depths. How-

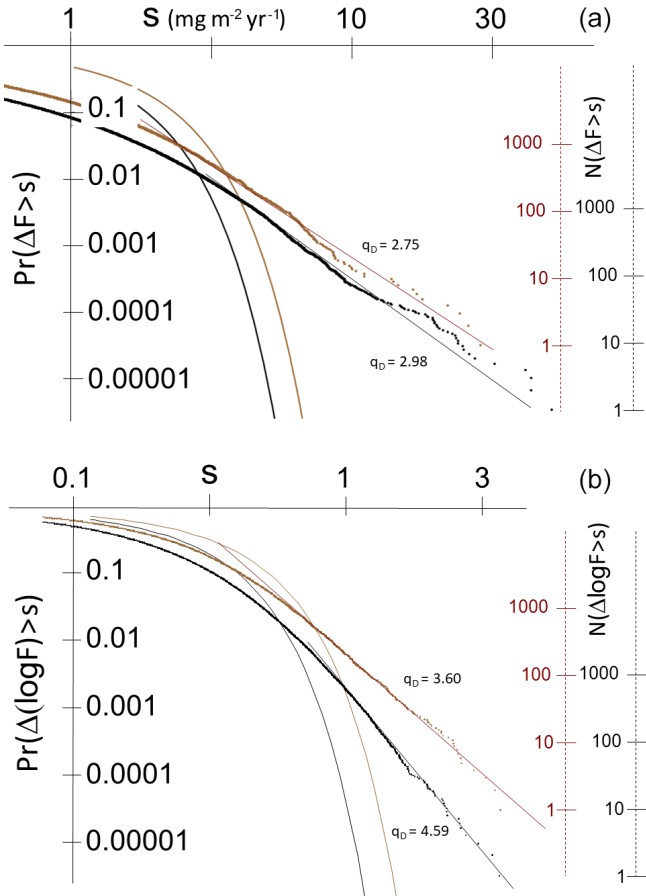

**Figure 6. (a)** The probability distribution $Pr(\Delta F > s)$ of random changes in dust flux ($\Delta F$) exceeding a fixed threshold $s$ in time at a 25-year resolution (brown; 32 000 points) and in depth at 1 cm resolution (black; 251 075 points corresponding to the last 400 kyr). The frequency scales on the right give the number ($N$) of jumps in each of the series that exceeds the threshold $s$. The straight lines indicate power law probability tails with exponents $q_D$ indicated. Also shown (parabolas) are the Gaussians with the same mean and standard deviations. In time, the maximum change in flux corresponds to about 28 standard deviations (i.e. to a Gauss probability $\approx 10^{-91}$); in depth, it corresponds to 51 standard deviations (i.e. to $p \approx 10^{-455}$). On the right, we provide axes giving the actual number of flux increments that exceed $s$ (brown for the fluctuations in time; black for those in depth). **(b)** Same as **(a)** except for the increments of the log of the dust flux (brown is in time, 25-year resolution; black is in depth, 1 cm resolution). The curves are the closest-fitting (log-) Gaussians. The threshold $S$ is dimensionless, and the numerical values are correct if $F$ is measured in units of milligrams per square metre per year.

ever, we prefer using the fluxes because they are more directly representative of climatic changes than concentrations.

However there are some results that are worth mentioning. For example, Lovejoy and Schertzer (2012) performed a numerical analysis of the uncertainties in first- and second-order exponent estimates obtained from Haar fluctuations of

a universal multifractal model with $C_1 = 0.1$ and a range of values of $H$ (close to the value found here; see Figs. 9, 12). When the scaling ranges covered factors of about 1000, they found only a small bias ($\approx 0.02$) in estimates of $H$ and a comparable uncertainty. However, in practice – such as the estimates here – the main source of uncertainty is the subjective choice of the scaling range itself: Fig. 9 shows that values of slopes depend on the region over which trends are fit, hence the straight reference lines.

Finally, for the problem of estimating probability tail exponents ($q_D$; Fig. 6a, b), Clauset et al. (2009) found that the maximum likelihood method is optimal. However, they assumed that the range over which the power tail was valid was pre-determined. The real difficulty in Fig. 6a and b is that one must make an initial subjective choice about the exact range over which the exponent is estimated; using sophisticated estimators does not seem warranted.

## 3.4 Phases

Scaling is a statistical symmetry, a consequence of a time and space scaling symmetry of the underlying dynamics. Being statistical means that *on average* the statistics at small, medium and large scales are the same in some way (more precisely, it holds over a statistical ensemble). The difficulty is that on a single realization – such as that available here, i.e. a single core from a single planet earth – the symmetry will necessarily be broken. For example, in the spectrum in Fig. 4, in each of the proposed scaling regimes, scaling only predicts that the actual spectrum from this single core will vary about the indicated straight lines that represent the ensemble behaviour. Since this variability is strong, we made the potential scaling regimes more obvious by either averaging the spectrum over frequency bins (the red and blue spectra) or by breaking the series into shorter parts and averaging the spectra over all the parts, effectively treating each segment as a separate realization of a single process (green). In any event, all that any empirical analysis can show is that the data are consistent with the scaling hypothesis.

This already illustrates the general problem: in order to obtain robust statistics we need to average over numerous realizations, and since here we have a single series, the best we can do is to break the series into disjointed segments and average the statistics over them, assuming that the major underlying processes were constant over the last 800 000 years. Yet at the same time, in order to see the wide-range scaling picture (which also helps to more accurately estimate the scaling properties or exponents), we need segments that are as long as possible. The compromise that we chose between numerous short segments and a small number of long ones was to break the series into eight glacial–interglacial cycles and each cycle into eight successive phases. As a first approximation, we defined eight successive 100 kyr periods (hereafter called "segments"; Fig. 7a), corresponding fairly closely to the main periodicity of the series. As we discussed,

the spectral peak is broad implying that the duration of each cycle is variable – the cycles are only "quasi-periodic". It is therefore of interest to consider an additional somewhat flexible definition of cycles as the period from one interglacial to the next (hereafter called "cycle"; Fig. 7b). The break points were taken at interglacial optima: 0.4, 128.5, 243.5, 336, 407.5, 490, 614, 700, and 789 kyr BP, i.e. $96.9 \pm 18.7$ kyr per cycle. Using the latter definition, the cycles were non-dimensionalized so that non-dimensional time was defined as the fraction of the cycle, effectively stretching or compressing the cycles by $\pm 19\%$.

With either of these definitions, we have eight segments or cycles, each with eight phases. Note that in our nomenclature, phases 1 and 8 are the youngest and oldest phases respectively and that time flows from phase 8 to phase 1. Figure 8 shows the phase-by-phase information summarized by the average flux over each cycle including the dispersion of each cycle about the mean (for the segments in panel a and the cycles in panel b). We see that the variability is highest in the middle of a cycle and lowest at the ends.

The spectra showed that there were wide-scale ranges that are on average scale–invariant power laws, and Fig. 4 quantifies the glacial–interglacial cycle CE5. We are thus interested in characterizing the scaling properties over the different phases of the cycle; for this we turn to real-space statistics. In Fig. 9 we compare the statistics averaged over cycles and the statistics averaged over phases. The figure shows that the phase-to-phase differences are much more important than the cycle-to-cycle differences. $\langle |(\Delta F(\Delta t))| \rangle$ (lower left). We could also note that since the different cycles had quite similar statistics (panels b and d), this implies that there is no bias in the flux estimates with depth of the core.

From the global statistics (e.g. Figs. 4, 5), it is clear that in each glacial–interglacial cycle there are two regimes, so that before characterizing the structure functions by their exponents (e.g. $H = \xi(1)$ for the mean fluctuations), we have to determine the macroweather–climate transition timescale $\tau_c$ whose average (from Figs. 4, 5) is 250–300 years.

One way of estimating the transition scale $\tau_c$ is to make a bilinear fit of $\log_{10} S_1(\Delta t)$ (i.e. Haar with $q = 1$, the mean absolute fluctuation) with the mean slopes $-0.05$ (small $\Delta t$) and slope $+0.35$ (large $\Delta t$; the values were chosen because they are roughly the $H$ estimates from the average over all the cycles) (Fig. 9). The hypothesis here was that there were two regimes, each characterized by a different exponent, each of which was estimated from the ensemble statistics. Therefore, the analysis only needed to estimate the scale at which the low-frequency process exceeded the high-frequency one. Bilinear fits were made for each phase of each segment (blue) as well as for each phase of each cycle (black). For each phase there were thus eight transition scales, which were used to calculate the mean and its standard deviation (shown here as representative black arrows). From the figure we see that at first (phases 8–3) the transition scale is relatively short (250–400 years) but that it rapidly moves to longer (1–2 kyr)

scales for the final phases 2 and 1. The average transition scale over all phases is around 300 years.

The figure shows that our results are robust since the results are not very different using dimensional and non-dimensional time (segments and cycles). Comparing the blue and black curves, we see that in all cases the late phases have much larger $\tau_c$ than the early and middle phases. Also shown in Fig. 10 (dashed) is a plot of the break points estimated by a more subjective method that attempts to visually determine a break point on $\log S_1$–$\log \Delta t$ plots. Again, we reach the same conclusion with quantitatively very similar results: a transition of millennia for phases 1 and 2 and a few centuries in the middle of the cycle. The cycle average value ($\tau_c \approx 300$ years) is therefore not representative of the latest phases where $\tau_c$ is many times larger (glacial maxima and interglacials). The Holocene has an even larger transition scale ($\tau_c = 7.9$ kyr, marked by an X in Fig. 10), but it lies just outside the standard deviation of the first non-dimensional phases (red arrows in Fig. 10). Although the Holocene value of $\tau_c$ is the largest in phase 1, it corresponds to 1.55 standard deviations above the mean with (assuming a Gaussian variability) a corresponding $p$ value of 0.12, roughly the expected extreme of a sample of eight; it is therefore not a statistical outlier.

Alternatively, rather than fixing a phase and determining the variation in the mean fluctuation and intermittency function (Fig. 9), we can consider the variation in the Haar fluctuations at fixed timescales and see how they vary from phase to phase (Fig. 11). The figure shows the phase-to-phase variation in Haar fluctuations at 50, 100, 200, 400, 800, 3300, and 7000 years scales (bottom to top; the dashed and solid lines alternate to demarcate the different curves; they are not uncertainties). Over the macroweather regime (up to about 400–800 years), the fluctuations tend to cancel so that the variability is nearly independent of timescale. In contrast, once we reach the longer scales in the climate regime (up to 7000 years), the fluctuations increase noticeably as the time interval $\Delta t$ is increased. For every timescale, there is a clear cyclicity (left to right), with fluctuation amplitudes largest in the middle phases. We note that the cycle-to-cycle variability is fairly large; about a factor of 2 (for clarity the error bars indicating this cycle-to-cycle spread were not shown).

Finally, we describe for each phase the drift tendency and the intermittency as well as fluctuation amplitude and extremeness of the data. In Fig. 12 we show the result on the non-dimensional phases of the range 500 years $< \Delta t <$ 3000 years (panels a and b CE6; the range was chosen to be mostly in the climate regime, i.e. with $\Delta t > \tau_c$, and it was fixed so as to avoid any uncertainty associated with the algorithm used to estimate $\tau_c$). Recall that the fluctuation exponent $H > 0$ quantifies the rate at which the average fluctuations increase with timescale. Similarly, the exponent $C_1$ characterizes the rate at which the spikiness near the mean (the intermittency exponent) increases with scale. We see (panel a) that $H$ is fairly high in the early phases with $H$ reaching small value in the later phases (with $H$ actually a

Please note the remarks at the end of the manuscript.

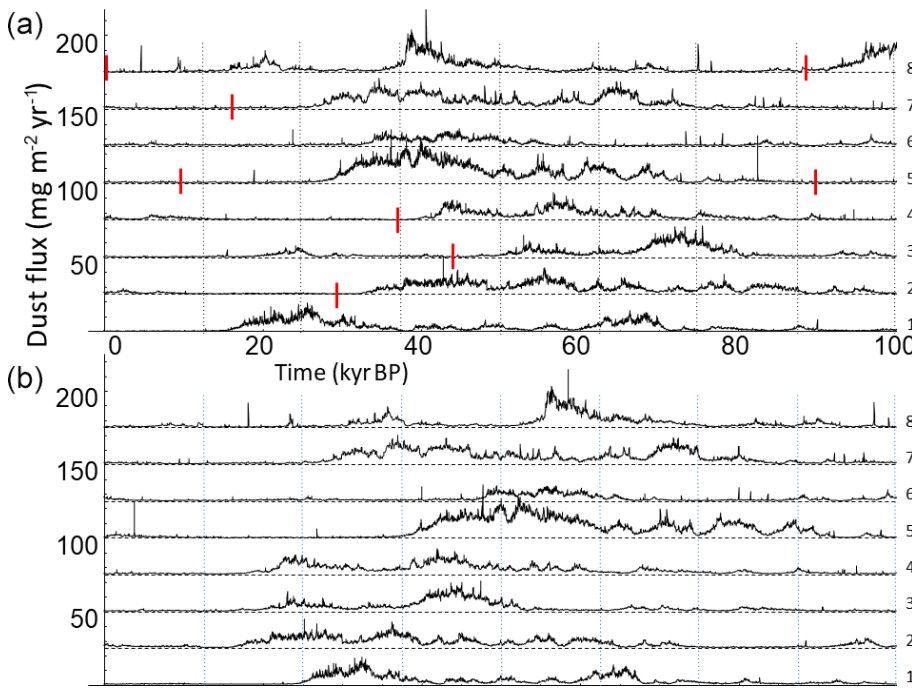

**Figure 7.** Panel **(a)**: successive segments of theoretical 100 kyr long glacial cycles using usual (dimensional) time (present to past: bottom to top, the segment number is at the far right) with the 12.5 kyr phases indicated by vertical dashed lines. The short red lines indicate the interglacial dust minima. Each glacial–interglacial cycle is shifted by 25 units in the vertical for clarity. The red markers in **(a)** are mapped to the first dashed blue line in **(b)**. Panel **(b)**: successive cycles using non-dimensional time (interglacial to interglacial) and then shifted by one phase to better line up with the usual time segments (the left-most phase of the bottom line of **(b)** is zeroed). The average (nominal) resolution is 25 years. The interglacial dust minima were taken as 128.5, 243.5, 336, 407.5, 490, 614, 700, and 789 kyr BP, and the data start at 373 yr BP. Each cycle is shifted by 25 units in the vertical for clarity. The data older than 789 kyr BP were not used in these non-dimensional cycles.

little bit negative on average in phase 1 due to the large $\tau_c$ value). $C_1$, on the other hand (panel b), decreases a bit in the middle the phases. The error bars show that there is quite a lot of cycle-to-cycle variability.

If $H$ quantifies the "drift" and $C_1$ the "spikiness", then Fig. 12 shows that the early phases have high drift and medium spikiness, and the middle phases have high drift and lower spikiness, while phases 1–2 have low drift but medium spikiness. To understand this better, consider the transition timescales in Fig. 10. The youngest two phases with the low drift and spikiness are also the phase with the longest transition scales. This means that the rate at which the variability builds up is small and that it only builds up over a short range of scales (from $\tau_c$ to roughly $\Delta t = 50$ kyr – the half cycle duration; this can be checked in Fig. 9, which shows the phase-by-phase structure functions and intermittency functions). Conversely, phases 3 and 4 with high drift and high intermittency also have a smaller $\tau_c$ so that both the fluctuations and spikiness build up faster (Fig. 11) and over a wider range of scales (Fig. 10).

Another useful characterization of the phases is to directly consider the flux variability at a fixed reference scale, taken here as the 25-year resolution; quantifying the amplitude of the variability of each segment by its standard deviation $A$ at 25-year timescale (Fig. 12c). This is not the *difference* between neighbouring values or fluctuations (as in Fig. 11), it is rather the variability of the series itself at a 25-year resolution. For each of the phases, we have eight estimates (one from each cycle); these are used to calculate the mean (central solid line) and standard deviation shown by the error bars showing the cycle-to-cycle dispersion of the values. We can see that the amplitude of the 25-year-scale fluctuations is about 4 times higher in the middle of the ice age (phase 4) than at the interglacial (phase 1). The figure clearly shows the strong change in variability across the cycle.

Whereas $C_1$ characterizes the intermittency near the mean, we have seen that the probability exponent $q_D$ characterizes the extreme spikiness. An extreme (low) exponent $q_D$ phase implies that most of the time the changes in flux are small, but occasionally there are huge transitions. Conversely, a high (less extreme) $q_D$ implies that there is a wider range of different flux changes so that most of the changes tend to be in a restricted range. Figure 12d compares $q_D$ phase by phase. If the value of $q_D$ is smaller, the extreme fluctuations are more and more extreme relative to the typical ones. Therefore, from the figure, we see that the extremes are stronger in the be-

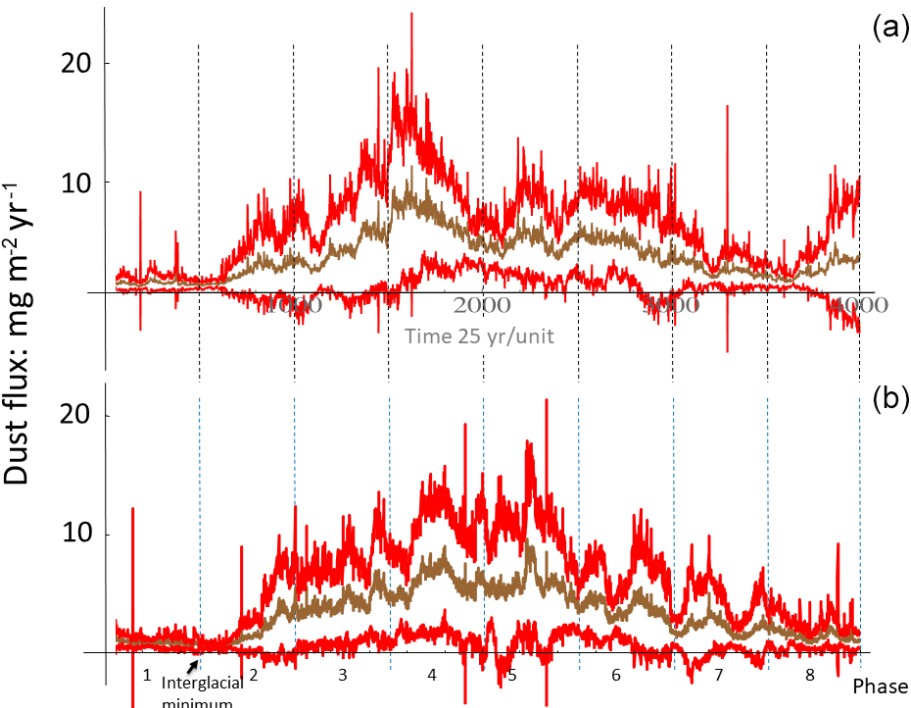

**Figure 8.** Panel **(a)**: averaging over the eight cycles at a 25-year resolution, we get the above picture: the mean is brown and the 1 standard deviation cycle-to-cycle variability is shown in red. The dashed vertical lines give a further division into $8 \times 12.5$ kyr segments, the eight "phases" of the cycle. Panel **(b)**: the same but for the non-dimensional time. The relative position of the interglacial minimum at the first dashed line is indicated.

ginning and end of the cycle and somewhat less pronounced in the middle phases of the cycle (note that the overall mean is $2.62 \pm 0.42$; this can be compared to the value $q_D = 3.60$ for the overall log-transformed data; Fig. 6b). Notice that for phase 8, $q_D = 2.03$; this is close to the value $q_D = 2$, below which the extremes are so strong that the variance (and hence spectrum) does not converge. Summarizing, we can now categorize the phase-by-phase spikiness as strong extremes and medium spikiness (phases 1, 2, 8) and intermediate extremes and low spikiness (phases 3–7). For the cycle-to-cycle estimates (not shown), the value $q_D = 2.75 \pm 0.41$ seems to be fairly representative of all the cycles, although there is a slight tendency for $q_D$ to decrease for the older cycles implying that they may have been a bit more extreme than the recent ones.

## 4 Discussion

An attractive aspect of dust fluxes is that they are paleo-indicators with unparalleled resolutions over huge ranges of temporal scales. However, they come with two difficulties. First, their dynamical interpretation is not unambiguous: they depend on temperature, wind, and precipitation; dust flux variability is hard to attribute to a specific process, and it is a holistic climate indicator. Second, their appearance as a sequence of strong spikes is unlike that of any of the fa-

miliar proxies. Indeed, we argue that their highly spiky (intermittent) nature (i.e. with $C_1 > 0$) is outside the purview of conventional statistical frameworks including autoregressive, moving average, or more generally quasi-Gaussian or even quasi-log-Gaussian processes.

Due to the dominance of the continuum (spectral background) variability, physical interpretations must be based on an understanding of climate variability as a function of scale. We first consider overall analyses over the whole dust flux series and then focus on the phases. The spectral analysis (Fig. 4) is the most familiar, and for the dust fluxes, it is qualitatively similar to previous results obtained with temperature data, although temperature spectra with anything approaching the resolution of Fig. 4 are only possible over the most recent glacial cycle. The most striking spectral feature is the peak over the background at 100 kyr periodicity. The broadness of this peak already indicates the irregularity of the Earth system response to the eccentricity-forced orbital cycles. The (near-) absence of obliquity frequencies at 41 kyr is notable and is consistent with the corresponding analysis of paleotemperatures. Although there is definitely power in that frequency range, it is barely larger than the background continuum, suggesting a low response to that forcing. Finally, our high-resolution data allow us to discern two different power law regimes: one at low frequencies with an exponent

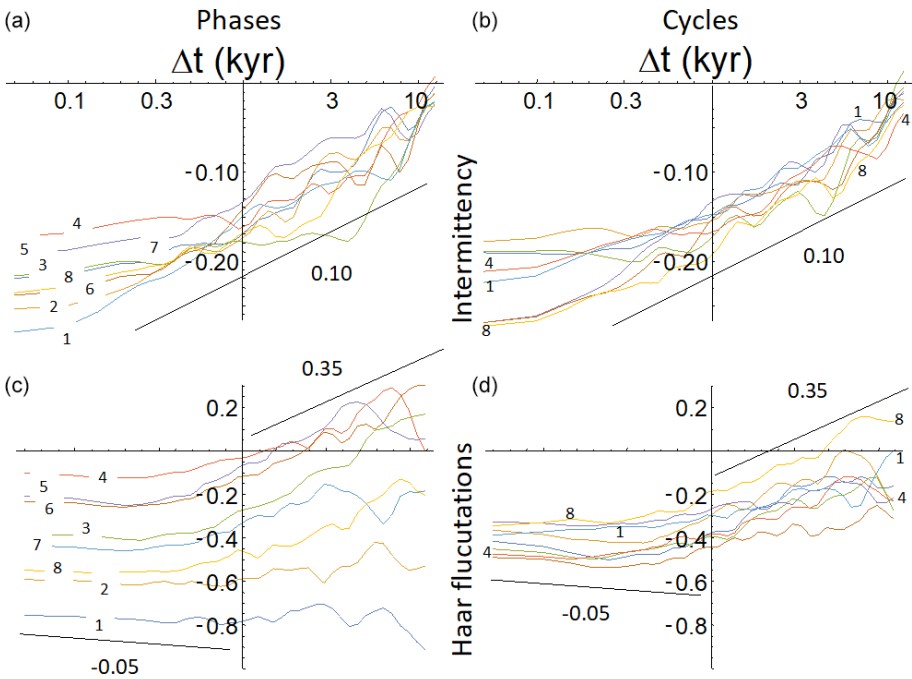

**Figure 9.** Panels **(a, b)** show the intermittency function $G(\Delta t)$ (whose slope on the log–log plot is $C_1$), and panels **(c, d)** show the mean absolute Haar fluctuation $S_1(\Delta t)$ (whose slope on the log–log plot is $H$); **(a, c)** show the result for each phase after averaging over the eight cycles with the numbers next to each line indicate=ing the phase number (each colour corresponds to the same number); **(b, d)** show the result for each cycle after averaging over the phases. Here, the same colours and numbers correspond to the cycle number; shown are only cycles 1, 4, and 8 to avoid clutter. Whereas each cycle is fairly similar to every other cycle (the **(b, d)**), each phase is quite different (**(a, c)**). We see that the most significant difference is the fluctuation amplitude as a function of phase (**(c)**).

$\beta = 1.7$ and one at high frequencies with exponent $\beta = 0.8$, with the transition between the two at around 300 years.

In Sect. 2.3, we discussed some of the difficulties inherent in interpreting spectra and showed that the exponent of the integrated spectrum $\beta - 1$ is more directly relevant than $\beta$ (ignoring intermittency, this is the same as the wandering or cancelling criterion $H > 0$ or $H < 0$). Applying this understanding to the dust exponents, we see that in macroweather, there is a weak high-frequency dominance ($1 - \beta \approx 0.2 > 0$), whereas the climate regime is dominated by low frequencies ($1 - \beta \approx -0.7 < 0$). A plausible physical explanation is that over long periods of time (at climate regime scales), the amount of dust in the SH CE7 atmosphere is driven by changes in glacier and vegetation coverage, which is itself forced by SH temperature change. There is therefore a very strong correlation between dust and temperature at climatic scales (Lambert et al., 2008). At higher (macroweather) frequencies, temperature oscillations are too fast to overcome the inertia of ice sheet and vegetation responses; dust and temperature correlations are very low. Instead, dust deposition in Antarctica will be more sensitive to temporary atmospheric disturbances in the winds and the hydrological cycle.

To interpret the analysis by the phase of the dust record (Fig. 12), one must understand the significance of $A$ and of the exponents $H$, $C_1$, and $q_D$ in the context of dust deposi-

tion. The $H$ exponent and the amplitude $A$ are directly linked to mean fluctuations and values, $A$ being the standard deviation ($\langle F^2 \rangle^{1/2}$) of the dust flux variability at a fixed (here, 25-year) timescale, whereas $H$ determines the rate at which the flux fluctuations ($(\langle \Delta F(\Delta t)^2 \rangle^{1/2})$) change with timescale $\Delta t$. We saw that a positive $H$ exponent signifies a tendency to drift, whereas when $H < 0$, the dust fluctuations tend to cancel each other out and the record will cluster around a mean value. In contrast, $H > 0$ indicates that the dust fluxes will not cluster around a mean value; in essence, the process wanders and does not stay constant; it appears to be unstable. The low $H$ numbers during phases 1 and 2 (interglacial and glacial maxima) indicate a very constant, stable climatic state, with Patagonian dust production being either very low during interglacials (low glacier activity, large vegetation cover) or very high (Patagonian ice cap fully grown, large outwash plains on the Argentinian side). In contrast, the high $H$ and amplitude $A$ values during the mid-glacial may have been due to strong variability in glacier extent during that time (García et al., 2018; Sugden et al., 2009) and therefore a very variable dust supply (see also Fig. 11 that shows how the amplitude of the fluctuations at different timescales varies with the phase). The glacial inception (phases 7 and 8) features low $A$ but a high $H$ exponent. This implies that the mean dust level was highly variable, but the dust supply was

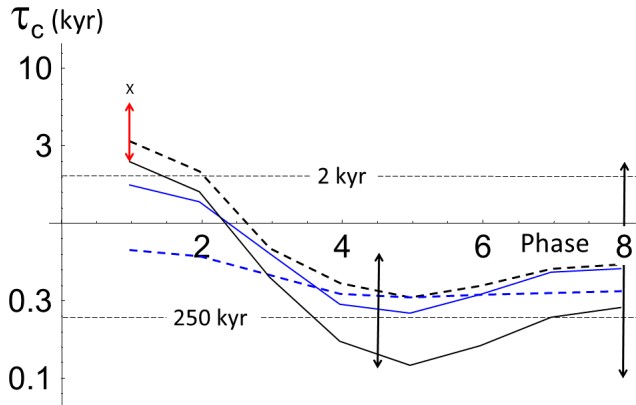

**Figure 10.** The transition scale $\tau_c$ estimated in two ways for each of the eight phases and from two definitions of the phases. The first method (solid lines) used a bilinear fit to the (logarithm) of the Haar $q = 1$ structure function (i.e. mean absolute fluctuation) as a function of log time lag $\Delta t$. To obtain robust results, a small $\Delta t$ region with the slope $-0.05$ and a large $\Delta t$ slope $+0.25$ was imposed with the transition point ($\tau_c$) determined by regression. This was done for each segment and cycle. For each phase there were thus eight transition scales, which were used to calculate the mean of the logarithm of $\tau_c$ and its standard deviation. Results are shown for dimensional (segments, blue) and non-dimensional time (cycles, black). The second method used to estimate $\tau_c$ was graphical and relied on a somewhat subjective fitting of scaling regimes and transitions, but without imposing small and large $\Delta t$ slopes (exponents $H$). The results are shown in dashed lines; they are quite similar, although we can note some differences for the first phase (dimensional, blue) and the middle phases (non-dimensional, black). There is also considerable cycle-to-cycle spread that was quantified by the standard deviations. In order to avoid clutter, typical spreads are shown by the double headed black arrows. Dashed horizontal lines show the ensemble mean transition scale (about 250 years) as well as ensemble mean for phases 1 and 2 (around 2 kyr), which stands out compared to the rest of the phases. The red arrow shows 1 standard deviation for the non-dimensional first phases, while the X marks the value of the Holocene $\tau_c$ (7.9 kyr) just outside the $1\sigma$ limit.

still low, thus not allowing for large amplitude fluctuations. The higher amplitudes in phases 6 and 7 indicate that dust supply became abundant then. Since the Argentinian continental shelf was still submerged at that moment and the out-
5 wash plains not yet fully extended, the higher dust emissions may have been due to a transformation in vegetation cover about 30 kyr after glacial inception, possibly accompanied by changes in glacial and periglacial processes in the Andes.

The exponents $C_1$ and $q_D$ are associated with the inter-
10 mittency or spikiness of the data. $C_1$ is a measure of the sparseness (or degree of clustering) of the mean-level spikes (i.e. whose amplitudes contribute most to the mean spike level). It is equal to one minus the fractal dimension of the set of spikes that exceed the mean level ($D_1 = 1 - C_1$).
15 $q_D$ characterizes how extreme the most extreme spike values are. The dust flux record is generally more intermittent

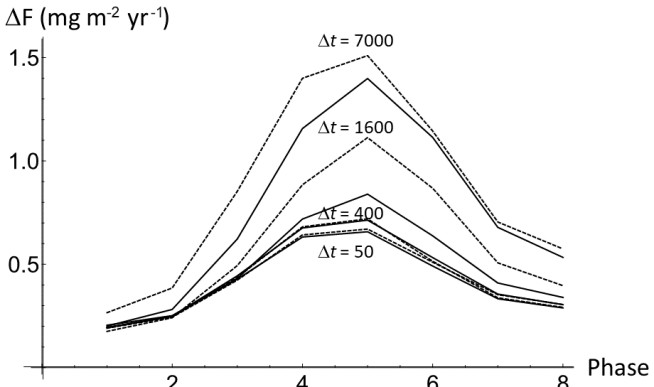

**Figure 11.** Using non-dimensional time, the amplitude of the Haar fluctuations are averaged over all the cycles The curves from bottom to top are for timescales of $\Delta t = 50$, 100, 200, 400, 800, 1600, 3500, and 7000 years, alternating solid and dashed lines (for clarity, only some of the $\Delta t$'s are marked). The cycle-to-cycle variability (the dispersion around each line) is about a factor of 2 (it is not shown to avoid clutter).

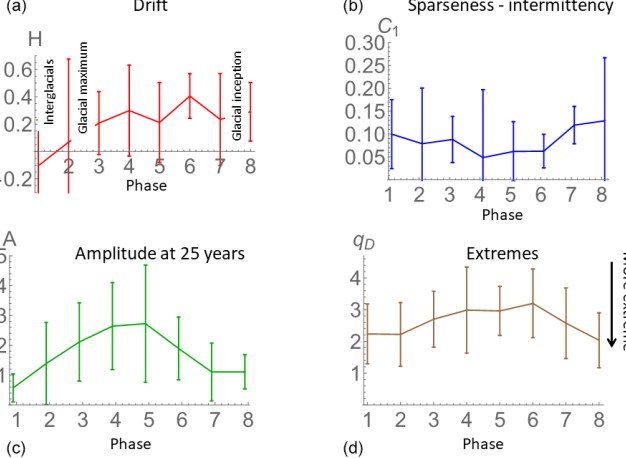

**Figure 12.** The fluctuation and intermittency exponents $H$ and $C_1$ ((**a, b**)) are estimated over the range 500–3000 years, as a function phase with the standard deviations from the cycle-to-cycle variability (all using non-dimensional time). Panel (**a**) ($H$) shows low drift in phases 1 and 2 but becomes driftier in the middle and older phases. The intermittency ($C_1$, (**b**)) is moderate at the beginning and end of the cycles and a little weaker in the middle. Panel (**c**) shows the amplitude of the fluctuations at 25 years determined by the standard deviation of the dust flux (units: milligrams per square metre per year). We see that the flux has low amplitude fluctuations at the beginning and end of the cycles and 3–4 times higher amplitude fluctuations in the middle. Panel (**d**) shows the probability exponent $q_D$ estimated from the 25-year resolution data for each phase; the extreme 5 % of the flux changes were used to determine the exponent in each phase; the cycle-to-cycle spread is indicated by the error bars (overall average over the phases: $q_D = 2.62 \pm 0.42$).

(with sparser, more clustered spikes, larger $C_1$) in phases 8, 1, and 2 (glacial inception, interglacial, glacial maximum) than in the mid-glacial, with also more extreme spike values (lower $q_D$). These power law fluctuations implied by the low values of $q_D$ are so large that according to the classical assumptions, they would be outliers. While Gaussians are mathematically convenient and can be justified when dealing with measurement errors, in atmospheric science thanks to scaling, non-linear dynamics, very few processes are Gaussian. This has important applications in tipping point analysis, where noise-induced tipping points are generally studied using well-behaved white or Gaussian noise (e.g. Dakos et al., 2012).

The exponents characterize the variability of the dust signal over a wide range of scales. To understand the two scaling regimes, it may be helpful to recall that the ice core dust signal depends on both the variability of the dust source and that of the overall climate system. For example, a spike in the dust source and a fast change in the system state (e.g. Dansgaard–Oeschger – DO – events in the NH) could both produce a similar signal. However, fast changes in system state – such as the DO events in the NH – apparently do not occur in the SH where the corresponding signals are more triangular and gradual in shape. High-frequency variations in dust deposition (at scales in the macroweather regime) are thus likely to be dominated by dust source dynamics rather than ice sheet changes that have generally larger reaction times. One hypothesis is that the transition timescale $\tau_c$ is the scale at which the source variability that decreases with scale ($H < 0$) becomes less than the system variability that increases with scale ($H > 0$). The macroweather variability is therefore likely dominated by vegetation and/or atmospheric changes. Large-scale natural fires could alter the landscape in a very short time, allowing for more dust uptake by the winds and a sudden rise in atmospheric dust. The recuperation of vegetation cover would be more gradual, though, resulting in a sawtooth shape of the dust spike that we do not observe in the data. Similarly, it has been suggested that rapid climate change in the Northern Hemisphere (e.g. DO events) would have synchronously changed the Southern Hemisphere atmospheric circulation and wind belts (Buizert et al., 2018; Markle et al., 2017). This could again have quickly changed the source or transport conditions but would again have resulted in a sawtooth-shaped peak, either by steady regrowth of vegetation in the dust source areas or as climate conditions in the north Atlantic gradually return to stadial (Pedro et al., 2018).

Finally, we could mention volcanoes. Volcano eruptions usually saturated the dust-measuring device and were mostly cut from the record. Using the sulfate record to identify eruptions is tricky because many large sulfate peaks do not have a corresponding dust peak. This means that even if you do have matching dust and sulfate peaks, it could be an eruption or a coincidence. Therefore, the influence of volcanic variability on the results cannot be completely eliminated, although our key results are fairly robust with respect to the phase of the cycle and are therefore unlikely to be influenced by volcanic eruptions.

Although the spikes occur at all scales (see Fig. 3), the most likely explanation for the (shorter) macroweather-scale dust spikes is disturbances in the atmosphere, involving either the winds or the hydrological cycle (or both at the same time). The obvious candidate for a perturbation that would lead to increased dust in the atmosphere is drought. We will therefore interpret macroweather dust spikes as multiannual to multidecadal or multicentennial drought events in southern South America. With this interpretation, we can conclude that glacial maxima, interglacials, and glacial inceptions were characterized by more frequent and more severe drought events than during the mid-glacial. During glacial maxima, such extreme dust events could have contributed to Southern Hemisphere deglaciation by significantly lowering ice sheet albedo at the beginning of the termination (Ganopolski and Calov, 2011). In contrast, more frequent dust events could have contributed to glacial inception through negative radiative forcing of the atmosphere.

## 5 Conclusions

Until now, a systematic comparison of the different glacial–interglacial cycles has been hindered by a limitation of the most common paleoclimate indicators – the low resolution of Pleistocene temperature reconstructions from ice or marine sediment cores. Due to this intrinsic characteristic, the older cycles are poorly discerned; we gave the example of EPICA paleotemperatures whose resolution in the most recent cycle was 25 times higher than the resolution in the oldest one. In this paper, we therefore took advantage of the unique EPICA Dome C dust flux dataset with 1 cm resolution measuring 320 000 cm, whose worst time resolution over the whole core is 25 years.

Dust fluxes are challenging not only because of their high resolutions, but also because of their unusually high spikiness (intermittency) and their extreme transitions that occur over huge ranges of timescales. Standard statistical methodologies are inappropriate for analysing such data. They typically assume exponential decorrelations (e.g. autoregressive or moving average processes) that have variability confined to narrow ranges of scale. In addition, they assume that the variability is quasi-Gaussian or at least that it can be reduced to quasi-Gaussian through a simple transformation of variables (e.g. by taking logarithms). In this paper, using standard spectral and probability distribution analysis, we show that both the spectral and the probability tails were power laws, not exponential and requiring nonstandard approaches.

The high resolution of the data allowed us to not only quantitatively compare glacial–interglacial cycles with each other, but also to subdivide each cycle into eight successive phases that could also be compared to one another. One of

the key findings was that there was a great deal of statistical similarity between the different cycles and that within each cycle there were systematic variations in the statistical properties with phase. These conclusions would not have been possible with the corresponding much lower-resolution temperature proxy data.

Our variability analysis using real-space (Haar) fluctuations confirmed that the majority of the variability was in the macroweather and climate scaling regime backgrounds with an average transition scale $\tau_c$ of about 300 years. In the climate regime (timescales above $\tau_c$), dust variability is more affected by long-term hemispheric-wide climate changes affecting slow-response subsystems like glaciers and vegetation, which explains the high correlation of dust and temperature at these scales. In contrast, dust variability in the macroweather regime (timescales below $\tau_c$) would have been more influenced by short-term atmospheric perturbations such as droughts and precipitation minima.

Using various techniques, $\tau_c$ was found to be systematically larger in the youngest two phases than in the middle and oldest phases; about 2 kyr but with nearly a factor of 4 cycle-to-cycle spread and equal to 300 years (with a factor of 2 spread) for the six remaining phases. For the Holocene, $\tau_c$ was found to be 7.9 kyr, which makes it an exceptionally stable interglacial, but not a statistical outlier compared to other interglacials. Similarly, the typical (rms) variation in flux amplitude was smaller in the early phase increases by (on average) a factor of 4 from $\pm 0.13$ to about $\pm 0.5\,\mathrm{mg\,m^{-2}\,yr^{-1}}$ in the middle and later phases. The Holocene (with an amplitude of $\pm 0.08\,\mathrm{mg\,m^{-2}\,yr^{-1}}$) was again particularly stable with respect to the phase 1 of other cycles, but it was not an outlier.

We addressed the task of statistically characterizing the cycles by primarily characterizing the phases' variability exponents $H$, $C_1$, $q_D$, and amplitude A. We show that the atmosphere was relatively stable during glacial maxima and interglacials, but highly variable during glacial inception and mid-glacial. However, the low amplitude of dust variability during glacial inceptions indicates that vegetation cover and dust production processes did not significantly change until $\sim 30$ kyr after glacial inception.

We interpret the intermittency indicators as suggesting a higher frequency of drought events and more severe droughts during glacial inception, interglacials, and glacial maxima than during mid-glacial conditions. These short-term spikes in atmospheric dust could have helped trigger Southern Hemisphere deglaciation through albedo feedback of ice sheet surfaces or glacial inception through negative radiative forcing.

The results presented in this paper are largely empirical characterizations of a relatively less known source of climate data: dust fluxes. Dust flux statistics defy standard models: they require new analysis techniques and better physical models for their explanation. These reasons explain why our results may appear to be rough and approximate. Readers may nevertheless wonder why we did not provide standard uncertainty estimates. But meaningful uncertainties can only be made with respect to a theory and we have become used to theories that are deterministic, whose uncertainty is parametric and that arises from measurement error. The present case is quite different: our basic theoretical framework is rather a stochastic one; it implicitly involves a stochastic "earth process" that produces an infinite number of statistically identical planet earths of which we only have access to a single ensemble member. Unfortunately, we do not yet have a good stochastic process model from which we can infer sampling errors and uncertainties. In addition, from this single realization, we neglected measurement errors and estimated various exponents that characterized the statistical variability over wide ranges of timescale, realizing that the exponents themselves are statistically variable from one realization to the next. In place of an uncertainty analysis, we therefore quantified the spread of the exponents (which themselves quantify variability). In the absence of a precise stochastic model we cannot do much better.

This paper is an early attempt to understand this unique very high-resolution dataset. In future work, we will extend our methodology to the EPICA paleotemperatures and to the scale-by-scale statistical relationship between the latter and the dust fluxes.

**Data availability.** The dust flux data are available here: https://doi.org/10.1594/PANGAEA.779311 (Lambert et al., 2012b) TS6 .CE8

**Author contributions.** .TS7

**Competing interests.** The authors declare that they have no conflict of interest.

**Acknowledgements.** Shaun Lovejoy's contribution to this fundamental research was unfunded. Fabrice Lambert acknowledges support by CONICYT projects Fondap 15110009, ACT1410, Fondecyt 1171773, and 1191223 and the Millennium Nucleus Paleoclimate. MN CE9 Paleoclimate is supported by the Millennium Scientific Initiative of the Ministry of Economy, Development and Tourism (Chile).

**Financial support.** This research has been supported by the CONICYT (FONDAP, project nos. 15110009 and ACT1410), and FONDECYT (project nos. 1171773 and 1191223), the Millennium Nucleus Paleoclimate, supported by CE10 the Millennium Scientific Initiative of the Ministry of Economy, Development and Tourism (Chile). TS8

**Review statement.** This paper was edited by Carlo Barbante and reviewed by Michel Crucifix and two anonymous referees.

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

## Remarks from the language copy-editor

CE1    Spelling adjusted to our house standards.

CE2    Please define.

CE3    Please define.

CE4    Does "little" here go with "using"? If so, it might be advisable to reword this sentence for clarity (for example, "not using the logarithm of the flux changes much" or similar). At the moment, it is somewhat unclear how it is supposed to be read.

CE5    Please confirm this change or clarify the original punctuation and structure.

CE6    Please confirm.

CE7    Please define.

CE8    Please review the content of the following sections carefully, as edits are not displayed in the track-changes PDF: "Financial support", "Data availability", and "Competing interests".

CE9    Does this stand for "Millennium Nucleus" or is this a different project/institution?

CE10    Please confirm.

## Remarks from the typesetter

TS1    Please confirm the inserted information.

TS2    The composition of Figs. 1–3, 5–9, and 11–12 has been adjusted to our standards. This also includes language adjustments to Figs. 1–5, 7–10, and 12.

TS3    Please provide the short title.

TS4    Please confirm d as a differential operator.

TS5    Please note that units have been changed to exponential format. Please check all instances.

TS6    Please note and confirm the inserted citation as well as the corresponding reference in the reference list.

TS7    Please note that the section "Author contributions" is mandatory.

TS8    Please note that the funding information has been added to this paper. Please check if it is correct. Please also double-check your acknowledgements to see whether repeated information can be removed or changed accordingly. Thanks.

TS9    Please note and confirm the updated reference.