# Peer review of "Spiky Fluctuations and Scaling in High-Resolution EPICA Ice Core Dust Fluxes"

_Climate of the Past, 2018_

## Referee Comment (RC1) · Michel Crucifix (Referee) · 20 Mar 2019

The present manuscript has undergone a first review round in Climate of the Past. Previous reviews had not formulated strong objections about the statistical treatment of the data, but were worried about poor presentation and lack of focus. The manuscript was rejected and resubmitted, which unfortunately resulted in depriving the current reviewers from the benefits of a point-by-point rebuttal to the first review. Upon inspection, most figures are the same, key equations are unchanged, but more room is given to the interpretation of the results, and the abstract is more informative and, to my opinion, better written.

Given these improvements, and given the fact that there is no major objection to the

interest of the statistical approach (see however specific comments below), the study should be published. This said, the authors may want to seize the opportunity of this review to clarify or perhaps improve a few zones of discomfort that, I think, has somewhat hindered the reception of their recent works.

First, something needs to be done about graphics. There are many problems. To cite but a few: vertical axes with different tick marks have been superimposed (Figures 4 and 5), labels show aberrant disproportions caused by vertical stretching (many figures), numbers alignment is sometimes inconsistent (horizontal axis on Figure 6), horizontal ticks seem to have been added by hand, with shadows (Figure 5), or inconsistently aligned (Figure 2), and colour legends are missing (Figure 9).

Second, it would be useful to have some estimates of the variance of estimators. Assuming a stationary signal, and given the amount of data at hand, which variance do we expect for the estimate of $\tau_c$, or $H$, or even $C_1$ ? Figure 10 seems indeed to make it clear that there is a significant difference between $\tau_c$ of phase 1 and 8; trends on Fig. 12 are much less obvious and having clearer ideas as to whether variations can be attributed to statistical sampling, and to non-stationarity (or at least, whether they are a sign that null-hypothesis of stationarity should be rejected), would be helpful.

I leave it to the editor whether addressing these two comments in full is a hard request, especially the second one. This brings me to the point by point comments:

- p. 2: "two extremes" : They are not "extremes". Perhaps write again: between daily and orbital time scales...

- p. 2 l. 30: "temperature record" → "deuterium record" (especially for readers of Climate of the Past, they know that deuterium concentration and temperature are not the same )

- p. 2 l. 34: "The analysis of the dust record" → "The dust record"

- p. 3, l. 10: "can themselves be power laws" : be more specific about the conditions that generate power laws

- p. 3, l. 18: "In particular, etc." this is not a sentence.

- p. 5, l. 8: "Since K(1)=0" add "by definition".

- p. 5, l. 8: "While the mean to RMS ratio is intuitive" : the sentence is unclear. In what sense is $C_1$ a 'mean to RMS' ?

- p. 5 Eq(9) : use standard notation $\lim_{\Delta q \to 0}$.

- p. 6, l. 13: full sentence needed after a semicolon ";"

- p. 8, l. 13: "The plot graphically conterposes two views of variability". I suspect you mean Figure 4, but yet what is meant by this sentence is still not clear to me.

- p. 8, l. 20: 100 ka is *not* a Milankovitch frequency (Milankovitch was unaware of 100-ka cycles, he focused on 40-ka cycles).

- p. 8, overall, I found the lines 17-30 difficult to read. Consider whether it is possible to express the same message more concisely. - p. 9, l. 5 "possibly (but not obviously) scaling". In order to be possible but not obvious, a strict definition of what scaling means is needed. - p. 10, l. 22: we compare, without 's' - p. 11, l. 13: "p value of 0.12". See the above comment. We do not really know whether the distribution is normal. Do we know anything about a theoretical distribution, or perhaps one that could be simulated ? - p. 13, l. 9: "low internal feedback" → "low response"
* * *

---

## Referee Comment (RC2) · Anonymous Referee #2 · 20 May 2019

By using non-standard approaches, the authors analyze in this paper the 320.000 cm-long EPICA Dome C dust flux record published in Lambert et al., 2012. I cannot judge on the statistical techniques adopted for characterizing the cycles – that I leave to expert reviewers in this (mathematical/statistical) field.

Glacial-interglacial cycles that are present in the EPICA record are subdivided into 8 phases showing systematic variations of their statistical properties. The interpretation of the variability of four key indicators (H, C1, $q_D$, A) provides some interesting paleoclimatic information. I have only some concern about the interpretation of A and H exponent and their link to the size of the Patagonian ice sheet (see below), as well as some minor comments/questions. If the statistical part is duly revised by an expert in the field, this paper is worth to be published in CP after some ***minor revisions***.

\*\*\*\*\*\*\*\*\*\*\*\*\*\*\*\*\*\*\*\*\*\*\*\*

**Page 2, lines 17 to 29**: they refer to figure 2 which is (according to the figure caption) redrawn from Lovejoy, 2017 – please reference to that paper in this paragraph.

**Page 6, lines 25-27:** if you consider every cm of core, also the dust record can be somewhat affected at depth. Is this something which should be mentioned here? Probably not. But keeping this in mind, please re-structure this first sentence and state that you just take published data from Lambert et al., 2012 and discuss them as they are.

**Page 7 , line 1:** dust CONCENTRATION measurements, please specify.

**Page 7, lines 1-4**: dust production depends on the source "intensity" that includes also the areal extent of the source which is variable depending on the exposed continental shelf.

**Page 7, Lines 5-9:** dust depositional flux variability is also related to the hydrological cycle at low frequencies.. and to temperature…this explains the high correlation between dust and stable isotopes in ice cores. Please restructure this sentence.

**Page 7, Lines 8-9:** "at high frequency dust deposition variability depends on wind and hydrological cycle": which is the reference for this assumption? Dust concentration/flux depends on the hydrological cycle at different timescales… And wind (transport) influences mostly size rather than concentration. I think the sentence " at high frequency dust deposition variability depends on wind and hydrological cycle" is more a conclusion of your study, as written in lines 17-20, page 13 "At higher frequencies…[…] …dust deposition in Antarctica will be more sensitive to temporary atmospheric disturbances in the winds and hydrological cycle"

**Page 7, Lines 9-10:** As above, the sentence "..a single peak within a low background may revlect short-term atmospheric disturbance like drought over South America or low precipitation over the S.Ocean…" is more a conclusion of your work rather than a literature assumption. But in any case, why not an eruption? Why not an impurity (contamination) within the core? And at depth, why not a level where particles aggregates are present and to some extent perturb the signal? Is it certain that every spike registered in the core represents a climatic signal? It would be a huge work to analyze every sample where dust levels are above background, but I feel confident that many of these spikes can be attributed to these causes.

**Page 10, line 22:** replace "compares" with "compare"

**From Page 11, line 31, to page 12, line 24:** the whole paragraph is very interesting as figure 12 is also interesting. But with so many acronyms or indices, would it be possible to write for example "DRIFT" over the first plot (H), "SPIKINESS" over the second (C1) …etc? And also, maybe draw a horizontal arrow in each plot, going from right to left with "TIME" written on it. And why not, close to each number 1,2,3… the informal name of the phase ("interglacial", "glacial maximum",….)? This just for clarification, and for helping this figure to give an immediate message to the reader; I think this is one of the most important figures in the paper, so put it into value.

**Page 12, lines 26-29:** I would not say that the precise climate significance of dust flux is hard to nail down. Rather, maybe you can find an elegant way to say that dust fluxes result from several synergic variables and dust flux alone does not allow distinguishing the contribution of each of these variables in detail.

**Page 13, lines 5-8:** is the broadness of the peak really indicating irregularities of the eccentricity-forced Milankovitch cycles or, as I think, you probably mean it is indicating the irregularities in the continental response, including sea level change and shelf exposure, vegetation, glacial activity…and so on?

**Page 13 lines 15 to 20**: this is an important consideration and conclusion of this paper that needs to be hemphasized a bit more.

**Page 14, lines 1-2:** after saying on page 12 that it is complicate to associate the dust flux increase or decrease to one variable, you are now associating high dust supply during phases 6-7 to the size of the Patagonian ice cap.
 That is not correct, first of all because dust influx to Antarctica does not depend solely on source production. Yet, even considering only dust availability at the source (source production), and only solely

glacial dust sources, then you must consider that dust production is not one-to-one related to the size of the Patagonian ice sheet, but to the intensity of all glacial and periglacial processes potentially involved in dust production, which change in time, of course. Therefore, not only glacial processes related to the size of the ice sheet (involving  erosional processes related to the movement of ice and pressure on the underlying surface leading to great amounts of erosion)  are involved.  Also transport and deposition of glacial debris and formation of tills, outwash sediments (glacio-fluvial process), glaciolacustrine deposits, glacioeolian deposits,  can act as dust sources; in addition, dust can derive also from periglacial processes related to nivation, frost action, mass wasting, fluvial processes and eolian processes that are enhanced by freeze drying of surface sediments, scarce vegetation cover and exposure to strong winds.  So I think it is too simplistic to relate the A and H exponent to the size of the Patagonian ice sheet... please consider the possibility to relate these indices to the intensity of glacial and periglacial processes in South America.

About interpretation of C1 and qD exponents,  related to short-term events. You cite possible short-term disturbances in the atmosphere. Why you do not consider volcanic eruptions? Probably because of the short-term atmospheric disturbance of these events? Or because you do not have a corresponding sulphate signal in the core? Please clarify introducing one or more sentences before the conclusion paragraph.

---

## Referee Comment (RC3) · Anonymous Referee #3 · 22 May 2019

Review for "Spiky Fluctuations and Scaling in High-Resolution EPICA Ice Core Dust Fluxes" by Lovejoy and Lambert

In their paper Lovejoy and Lambert present an application of fluctuation analysis to a high-resolution reconstruction of dust fluxes in central Antarctica over the last 800 ka. This type of analysis with a record that is this highly resolved is new and generally merits the publication of the manuscript.

That being said, there are a number some minor and a number of major concerns that need to be addressed before the manuscript can be published in Climate of the Past.

I cannot speak to the correctness of the statistical analysis as I am not an expert on fluctuation analysis. That being said, I can probably speak from the point of view of

a large fraction of the potential readers of a Climate of the Past paper: I found the description of the method reasonably approachable. Personally, I would have liked to get more intuition for the method, its results and its possible applicability to other paleoclimate records.

Overall, the paper could be improved greatly by adding more discussion of the methods, their results and their interpretation. At present the paper focuses a lot on the listing of the results of the different statistical analysis. This takes away from the potential interest of both the method and the results to the wider paleo-climate community.

One major concern is, that the manuscript is lacking a clear description of the data set that has been used, the way it was generated, and how this affects the analysis presented here. I know that this description is given in the original publication. Nevertheless, I think this is a vital point given that Lambert et al. (2012) state to keep the generation of the dust flux reconstruction in mind when interpreting its variance as it is affected by the assumptions and corrections that were involved.

One other major concern that I have is, that the interpretation of the, I think interesting, results unfortunately seems to be ad-hoc and not very thoroughly argued. From my reading of the previously rejected version of this paper and its reviews this point has only been improved marginally. To strengthen the manuscript and make it more suitable for Climate of the Past, I hope the authors extent the discussion of the results both in comparison with other studies and in terms of their paleo-climatological interpretation.

The discussion and the results sections are completely lacking any information on the uncertainties of the obtained results. The large variability of the results between the different glacial/interglacial cycles indicates to me, that the results might not be very robust. One further observation that I made is that in many Figures the authors omit error or variability indications "for the sake of clarity" which I think is a poor choice. Additionally, the analysis is hinged upon a number of assumptions that are not justified in the text. Specifically, the slopes used for the breakpoint analysis and the range of

time scales used for the fluctuation analysis. The influence of these choices on the results needs to be shown and clearly discussed.

Specific comments:

Abstract:

P1 L17: The dataset has only a maximum resolution of 5 years. How can fluctuations on the one-year time scale be resolved. Please rephrase.

P1 L24-27: The logic of this sentence is not clear. Please rephrase.

P1 L27f: Why do they suggest this?

Introduction:

P2 L12-15: Please state that you refer to the temperature proxy time series, and also mention that this is a proxy, not a temperature in the strict sense.

P2 L23-27: Please consider explaining why the macro weather to climate transition timescale is important.

Method:

P3 L15: . . .the spectrum is the Fourier transform. . .

P3 L24-26: It is unclear why due to scale invariance, the results from the dust fluxes can be transferred to the temperature proxies if they are affected by different climatic mechanisms.

P3 L26: The analysis in the "future publication" should more thoroughly be discussed here, especially as you present results from it, or alternatively only be mentioned in the outlook. In my experience these "future publications" unfortunately often do not manifest themselves.

P4 L 24: extra comma between "compare" and "the"

Results:

The data set description should be in the Data + Methods section and more time should be spent describing the dataset used as pointed out above

P25 L27: With the strong emphasis that is put on the number of datapoint and their sampling frequency throughout the manuscript, the numbers should add up. Generally, this information is not strictly necessary for the paper and could be safely removed. Especially in light of the fact that the dust flux reconstruction is a combination of a multitude of measurements and ice flow modelling.

P7 L1: Water isotopes cannot be assigned to one particular atmospheric variable either, even though they are often used to reconstruct Temperature. Please consider rephrasing

P7 L5ff: Please consider mentioning the recent publications by Markle et al (2018) and Schüpbach et al. (2018) that deal with this the relation of en-route washout and aerosol deposition on the ice sheets more quantitively.

P7 L19: There is no red line in the Figures.

P8 L7f: The unit of the spectral amplitude of the log-transformed fluxes is wrong.

P8 L10f: Consider moving the comparison of the spectral densities with the results of the fluctuation analysis to after their introduction or to the discussion section.

P8 L13ff: This list of scaling exponents is completely irrelevant here and should be removed for clarity.

P8 L14f: It is unclear why this supports the use of dust as a proxy for atmospheric variability. Please clarify.

P8 L 25-30: The results presented here are very hard to follow. Please consider reformulating.

P9 L1: Whether the Haar fluctuations of the dust flux have simple interpretations is not shown in the Figure but is rather a matter of taste and should be left to the reader to decide. Please reformulate or remove this statement.

P9 L6f: After spending a lot of time describing and interpreting the spectral analysis in the previous section the description here is rather short. As the fluctuation analysis is the main focus of this study the authors should spent more time describing their results and leading the uninitiated reader through them.

P9 L4: "positive definite" seems to be the wrong phase here, consider replacing with "always positive" or similar.

P9 L18: Why is it important that the value is similar to those obtained from other ice cores. Please move all comparisons to other studies to the discussion clarify the interpretation and relevance.

P9 L20: This statement is somewhat superfluous: If the dust fluxes are not log-normally distributed due to the occurrence of large spikes, their logarithm will not be normally distributed.

P9 L24: Please move the interpretation and the relevance for tipping point analysis to the discussion and consider expanding on this point if you think it is an important application.

P9 L30f: Strictly, the statement that the scaling spectrum is the underlying behavior is an assumption and has not been shown in this study. Even though this a reasonable assumption, consider rephrasing.

P10 L4f: Please clearly state that any of the stacking approaches assume that all the glacial cycles and their sub phases are realizations of the same underlying process.

P10 L20ff: The start of this paragraph makes the reader expect spectra averaged over the different cycles. Please consider rephrasing and more extensively introducing the Figure.

P10 L29ff: The Figure indicates a slope of 0.35, not 0.25 as mentioned in the text. More importantly it is entirely unclear why the authors choose to set the slopes before and after the transition as constants and then only fit the transition time scale. The effect of the chosen values on the presented results needs to be clearly discussed and the uncertainty of the results are missing completely from the text. I strongly urge the authors to do a proper breakpoint and error analysis.

P11 L5-9: As the other method is arguably only slightly more objective than the breakpoint inference by hand, please consider removing this entire section.

P11 L20f: I suggest that the authors remove a couple of lines from the figure for clarity instead of the uncertainties or find a different way of visualizing the results.

P11 L23: Please state why this exact time range was chosen.

P12 L10: There is nothing black in Figure 12, please correct.

P12 L18-21: Consider moving this sentence to the beginning of the paragraph to make it easier for the reader to follow the results.

Discussion:

P13 L7-9: You do not perform any significance analysis, so please reformulate this statement. Also please discuss why the lack of power around the obliquity cycle is surprising.

P13 L28-32: Is there any supporting evidence for the glacier variability?

P14 L1: If A reflects the amplitude of the variance a change indicates only a change in the variability, not in their overall abundance, please reformulate.

P14 L14-17: Please discuss this statement in the context of recent proxy and model studies that indicate fast Southern Hemisphere circulation changes during DO events. (Markle et al. 2016, Buizert et al. 2018, Pedro et al. 2018).

P14 L24: Only ice cores have the intrinsic property that they become less resolved with increasing depth and thus age. Sediment cores are not affected by this. Please reformulate.

P14 L30: The paper does not show that the data neither over-samples nor smoothes. Please either add this to the paper or remove this statement.

P15 L24: The reduction to the characterization of the different phase is a decision that the authors take, not an intrinsic property or the result of some analysis. Please reformulate this statement.

P15 L25: Missing dot between "A" and "We"

P15 L 26f: How is the variability of the dust flux at Dome C connected to the "activity" of the Patagonian ice sheet. Please extent and clarify this argument.

Figures:

6b: The units for the axis are wrong.

11: I suggest that the authors remove a couple of lines from the figure for clarity instead of the uncertainties or find a different way of visualizing the results.

Cited literature:

Buizert, C. et al. (2018), Abrupt ice-age shifts in southern westerly winds and Antarctic climate forced from the north, Nature, doi:10.1038/s41586-018-0727-5.

Lambert, F. et al. (2012), Centennial mineral dust variability in high-resolution ice core data from Dome C, Antarctica, Climate of the Past, doi:10.5194/cp-8-609-2012.

Markle, B. R. et al. (2016), Global atmospheric teleconnections during Dansgaard-Oeschger events, Nature Geoscience, doi:10.1038/ngeo2848.

Markle, B. R. et al. (2018), Concomitant variability in high-latitude aerosols, water isotopes and the hydrologic cycle, Nature Geoscience, doi:10.1038/s41561-018-0210-

9.

Pedro, J. B. et al. (2018), Beyond the bipolar seesaw: Toward a process understanding of interhemispheric coupling, Quaternary Science Reviews, doi:10.1016/j.quascirev.2018.05.005.

Schüpbach, S. et al. (2018), Greenland records of aerosol source and atmospheric lifetime changes from the Eemian to the Holocene, Nature Communications, doi:10.1038/s41467-018-03924-3.

---

## Author Comment (AC1) · 19 Jun 2019

The present manuscript has undergone a first review round in Climate of the Past. Previous reviews had not formulated strong objections about the statistical treatment of the data, but were worried about poor presentation and lack of focus. The manuscript was rejected and resubmitted, which unfortunately resulted in depriving the current reviewers from the benefits of a point-by-point rebuttal to the first review. Upon inspection, most figures are the same, key equations are unchanged, but more room is given to the interpretation of the results, and the abstract is more informative and, to my opinion, better written. Given these improvements, and given the fact that there is no major objection to the interest of the statistical approach (see however specific comments below), the study should be published. This said, the authors may want

to seize the opportunity of this review to clarify or perhaps improve a few zones of discomfort that, I think, has somewhat hindered the reception of their recent works. First, something needs to be done about graphics. There are many problems. To cite but a few: vertical axes with different tick marks have been superimposed (Figures 4 and 5), labels show aberrant disproportions caused by vertical stretching (many figures), numbers alignment is sometimes inconsistent (horizontal axis on Figure 6), horizontal ticks seem to have been added by hand, with shadows (Figure 5), or inconsistently aligned (Figure 2), and colour legends are missing (Figure 9). Author Response: Thank you for the constructive comments. We have improved the figures according to your suggestions. Second, it would be useful to have some estimates of the variance of estimators. Assuming a stationary signal, and given the amount of data at hand, which variance do we expect for the estimate of $\tau c$, or H, or even C1 ? Figure 10 seems indeed to make it clear that there is a significant difference between $\tau c$ of phase 1 and 8; trends on Fig. 12 are much less obvious and having clearer ideas as to whether variations can be attributed to statistical sampling, and to non-stationarity (or at least, whether they are a sign that null-hypothesis of stationarity should be rejected), would be helpful. Author Response: It is actually not trivial to talk about the "variance" of the exponents, as the exponents themselves are only really meaningful in a stochastic framework: there is not a deterministic parametric model that underlies the distribution. As explained in a new paragraph (in the discussion section), to go further would require the elaboration of a more precise stochastic model that could predict the expected variation of exponents from one realization to another. We have rephrased the text to talk about dispersion instead. We also expanded the discussion of Figure 12 as you suggest. I leave it to the editor whether addressing these two comments in full is a hard request, especially the second one. This brings me to the point by point comments: • p. 2: "two extremes" : They are not "extremes". Perhaps write again: between daily and orbital time scales... Author Response: We changed "extremes" to "scales" • p. 2 l. 30: "temperature record" → "deuterium record" (especially for readers of Climate of the Past, they know

that deuterium concentration and temperature are not the same ) Author Response: Corrected • p. 2 l. 34: "The analysis of the dust record" → "The dust record" Author Response: Changed to "the dust data used here" • p. 3, l. 10: "can themselves be power laws" : be more specific about the conditions that generate power laws Author Response: Scale invariance of dynamics in time and in space lead to space-time statistics that are characterized by power laws. In this sentence we are referring to something rather different, to power laws in the tails of probability distributions. However, it turns out that several mechanisms have been proposed that link the two so that we could largely expect power law probability tails to emerge from space-time scaling dynamical processes. • p. 3, l. 18: "In particular, etc." this is not a sentence. Author Response: Corrected • p. 5, l. 8: "Since K(1)=0" add "by definition". Author Response: Changed to "Since $K(1) = 0$ is a basic property,…" • p. 5, l. 8: "While the mean to RMS ratio is intuitive" : the sentence is unclear. In what sense is C1 a 'mean to RMS' ? Author Response: We have clarified the sentence: "While the mean to RMS ratio is an intuitive statistic, it does not give a direct estimate of C1:…" • p. 5 Eq(9) : use standard notation $\lim \Delta q \to 0$. Author Response: We have modified the equation and text accordingly. • p. 6, l. 13: full sentence needed after a semicolon ";" Author Response: Corrected • p. 8, l. 13: "The plot graphically conterposes two views of variability". I suspect you mean Figure 4, but yet what is meant by this sentence is still not clear to me. Author Response: Changed to "The variability shown in Figure 4 can be interpreted broadly or in detail." • p. 8, l. 20: 100 ka is not a Milankovitch frequency (Milankovitch was unaware of 100-ka cycles, he focused on 40-ka cycles). Author Response: Changed to "orbital" throughout the text. • p. 8, overall, I found the lines 17-30 difficult to read. Consider whether it is possible to express the same message more concisely. Author Response: We tightened up the paragraph, it is shorter, hopefully it is clearer. – p. 9, l. 5 "possibly (but not obviously) scaling". In order to be possible but not obvious, a strict definition of what scaling means is needed. Author Response: We eliminate the "(but not obviously)". In this context, "scaling" means a power law behavior. Due to the shortness of the range and

the poor statistics at theses long time scales, the situation is not clear. - p. 10, l. 22: we compare, without 's' Author Response: Corrected - p. 11, l. 13: "p value of 0.12". See the above comment. We do not really know whether the distribution is normal. Do we know anything about a theoretical distribution, or perhaps one that could be simulated ? Author Response: The problem is that we don't have a detailed stochastic model. Without one, we don't know how much variability is "normal" or "typical". Using the standard deviation is agnostic in the sense that it is simply a standard nonparametric characterization of the spread. The only place where we go one step further is to assign a p value = 0.12. We have added a comment on this. - p. 13, l. 9: "low internal feedback" → "low response" Author Response: Corrected

Please also note the supplement to this comment:
https://www.clim-past-discuss.net/cp-2018-171/cp-2018-171-AC1-supplement.pdf

———————————————

---

## Author Comment (AC2) · 19 Jun 2019

By using non-standard approaches, the authors analyze in this paper the 320.000 cm-long EPICA Dome C dust flux record published in Lambert et al., 2012. I cannot judge on the statistical techniques adopted for characterizing the cycles – that I leave to expert reviewers in this (mathematical/statistical) field. Glacial-interglacial cycles that are present in the EPICA record are subdivided into 8 phases showing systematic variations of their statistical properties. The interpretation of the variability of four key indicators (H, C1, qD, A) provides some interesting paleoclimatic information. I have only some concern about the interpretation of A and H exponent and their link to the size of the Patagonian ice sheet (see below), as well as some minor comments/questions. If the statistical part is duly revised by an expert in the field, this paper is worth to be published in CP after some minor revisions. Author Response: Thank you for the positive comments.

*************************** Page 2, lines 17 to 29: they refer to figure 2 which is (according to the figure caption) redrawn from Lovejoy, 2017 – please reference to that paper in this paragraph. Author Response: Added Page 6, lines 25-27: if you consider every cm of core, also the dust record can be somewhat affected at depth. Is this something which should be mentioned here? Probably not. But keeping this in mind, please re-structure this first sentence and state that you just take published data from Lambert et al., 2012 and discuss them as they are. Author Response: The worsening of the resolution with depth of the dust data is discussed in lines 27-30. In our opinion, the sentence is clear that the dust data is not a product of this publication. Page 7 , line 1: dust CONCENTRATION measurements, please specify. Author Response: Changed to "dust flux measurements" Page 7, lines 1-4: dust production depends on the source "intensity" that includes also the areal extent of the source which is variable depending on the exposed continental shelf. Author Response: Original text changed to "The amount of dust deposited in East Antarctica will depend on the size and vegetation cover of the source region…" Page 7, Lines 5-9: dust depositional flux variability is also related to the hydrological cycle at low frequencies.. and to temperature…this explains the high correlation between dust and stable isotopes in ice cores. Please restructure this sentence. Page 7, Lines 8-9: "at high frequency dust deposition variability depends on wind and hydrological cycle": which is the reference for this assumption? Dust concentration/flux depends on the hydrological cycle at different timescales… And wind (transport) influences mostly size rather than concentration. I think the sentence " at high frequency dust deposition variability depends on wind and hydrological cycle" is more a conclusion of your study, as written in lines 17-20, page 13 "At higher frequencies…[…] …dust deposition in Antarctica will be more sensitive to temporary atmospheric disturbances in the winds and hydrological cycle" Page 7, Lines 9-10: As above, the sentence "..a single peak within a low background may reflect short-term atmospheric disturbance like drought over South America or

low precipitation over the S.Ocean. . ." is more a conclusion of your work rather than a literature assumption. But in any case, why not an eruption? Why not an impurity (contamination) within the core? And at depth, why not a level where particles aggregates are present and to some extent perturb the signal? Is it certain that every spike registered in the core represents a climatic signal? It would be a huge work to analyze every sample where dust levels are above background, but I feel confident that many of these spikes can be attributed to these causes. Author Response: Indeed, this sentence contains some results from our analyses. We have changed these sections to "High and low frequency variability in the dust flux record is likely driven by different processes. For examples, dust source conditions related to glaciers and vegetation cover may not have influenced high frequency variability due to their relatively slow rate of change. On the other hand, volcanic eruption or extreme events related to the hydrological cycle may produce high-frequency signals in the record." Indeed, analyzing every sample of the EPICA Dome C Continuous Flow Analysis data was a huge task and took F. Lambert over 2 years during his PhD. But thanks to this work we can be sure that most contamination peaks were cut out of the signal. Large volcanic eruptions usually saturated the dust signal and were cut out as well, but smaller more distant eruptions may produce a particle peak similar to a climatic event. Since we didn't check each dust peak for a corresponding sulphate peak, we have added in the text volcanoes as another possible high-frequency contributor. Aggregates did change the size distribution in the data quite a bit, but not the particle count. And since the dust flux record used here is a merged signal of particle concentrations and Ca and nssCa data, we feel confident that dust aggregates did not produce any sharp peaks. Page 10, line 22: replace "compares" with "compare" Author Response: Corrected From Page 11, line 31, to page 12, line 24: the whole paragraph is very interesting as figure 12 is also interesting. But with so many acronyms or indices, would it be possible to write for example "DRIFT" over the first plot (H), "SPIKINESS" over the second (C1) . . .etc? And also, maybe draw a horizontal arrow in each plot, going from right to left with "TIME" written on it. And why not, close to each number 1,2,3. . . the informal

name of the phase ("interglacial", "glacial maximum",….)? This just for clarification, and for helping this figure to give an immediate message to the reader; I think this is one of the most important figures in the paper, so put it into value. Author Response: We have followed the suggestions and updated the plot. Page 12, lines 26-29: I would not say that the precise climate significance of dust flux is hard to nail down. Rather, maybe you can find an elegant way to say that dust fluxes result from several synergic variables and dust flux alone does not allow distinguishing the contribution of each of these variables in detail. Author Response: We agree and changed this sentence to "First, their dynamical interpretation is not unambiguous: because they depend on temperature, wind, and precipitation, and so are holistic climate indicators, dust flux variability is hard to attribute to a specific process." Page 13, lines 5-8: is the broadness of the peak really indicating irregularities of the eccentricity-forced Milankovitch cycles or, as I think, you probably mean it is indicating the irregularities in the continental response, including sea level change and shelf exposure, vegetation, glacial activity…and so on? Author Response: Correct, we changed this to "The broadness of this peak already indicates the irregularity of the Earth system response to the eccentricity-forced orbital cycles." Page 13 lines 15 to 20: this is an important consideration and conclusion of this paper that needs to be hemphasized a bit more. Author Response: We expanded the corresponding section in the conclusion and added a sentence in the abstract. Page 14, lines 1-2: after saying on page 12 that it is complicated to associate the dust flux increase or decrease to one variable, you are now associating high dust supply during phases 6-7 to the size of the Patagonian ice cap. That is not correct, first of all because dust influx to Antarctica does not depend solely on source production. Yet, even considering only dust availability at the source (source production), and only solely glacial dust sources, then you must consider that dust production is not one-to-one related to the size of the Patagonian ice sheet, but to the intensity of all glacial and periglacial processes potentially involved in dust production, which change in time, of course. Therefore, not only glacial processes related to the size of the ice sheet (involving erosional processes related to the movement of

ice and pressure on the underlying surface leading to great amounts of erosion) are involved. Also transport and deposition of glacial debris and formation of tills, outwash sediments (glacio-fluvial process), glaciolacustrine deposits, glacioeolian deposits, can act as dust sources; in addition, dust can derive also from periglacial processes related to nivation, frost action, mass wasting, fluvial processes and eolian processes that are enhanced by freeze drying of surface sediments, scarce vegetation cover and exposure to strong winds. So I think it is too simplistic to relate the A and H exponent to the size of the Patagonian ice sheet... please consider the possibility to relate these indices to the intensity of glacial and periglacial processes in South America. Author Response: We agree that was too simplistic and we removed that passage from the abstract. We have changed that section to "The higher amplitudes in phases 6 and 7 indicates that dust supply became abundant then. Since the Argentinean continental shelf was still submerged at that moment and the outwash plains not yet fully extended, the higher dust emissions may have been due to a transformation in vegetation cover about 30 kyr after glacial inception, possibly accompanied by changes in glacial and periglacial processes in the Andes." About interpretation of C1 and qD exponents, related to short-term events. You cite possible short-term disturbances in the atmosphere. Why you do not consider volcanic eruptions? Probably because of the short-term atmospheric disturbance of these events? Or because you do not have a corresponding sulphate signal in the core? Please clarify introducing one or more sentences before the conclusion paragraph. Author Response: Identifying volcano eruptions using the sulphate record alone is tricky because many large sulphate peaks do not have a corresponding dust peak. This means that even if you do have matching dust and sulphate peaks, it could be an eruption or a coincidence. Unfortunately, tephras in the EDC ice core were measured only at very low resolutions to get an idea about eruption frequencies, and we do not have that data available to unequivocally identify single events. But you are right, this means we cannot exclude volcanoes. A paragraph was added explaining this. " "Finally, we could mention volcanoes. Volcano eruptions usually saturated the dust measuring device and were mostly cut

from the record. Using the sulphate record to identify eruptions is tricky because many large sulphate peaks do not have a corresponding dust peak. This means that even if you do have matching dust and sulphate peaks, it could be an eruption or a coincidence. Therefore, the influence of volcanic variability on the results cannot be completely eliminated, although our key results are fairly robust with respect to the phase of the cycle and are therefore unlikely to be influenced by volcanic eruptions."

Please also note the supplement to this comment:
https://www.clim-past-discuss.net/cp-2018-171/cp-2018-171-AC2-supplement.pdf

---

## Author Comment (AC3) · 19 Jun 2019

Shaun Lovejoy and Fabrice Lambert

lambert@uc.cl

In their paper Lovejoy and Lambert present an application of fluctuation analysis to a high-resolution reconstruction of dust fluxes in central Antarctica over the last 800 ka. This type of analysis with a record that is this highly resolved is new and generally merits the publication of the manuscript. That being said, there are a number some minor and a number of major concerns that need to be addressed before the manuscript can be published in Climate of the Past. I cannot speak to the correctness of the statistical analysis as I am not an expert on fluctuation analysis. That being said, I can probably speak from the point of view of a large fraction of the potential readers of a Climate of the Past paper: I found the description of the method reasonably approachable. Personally, I would have liked to get more intuition for the method, its results

and its possible applicability to other paleoclimate records. Author Response: Many thanks for the positive comment. Although the fluctuation analysis presented here has been previously used on some mostly modern climate records (temperatures, various climate indices, precipitation, CO2 concentrations etc.), none of these are as classically "paleo" or intermittent/spiky as the EDC dust fluxes. In the following publication we will make a comparison between various ice core dust and temperature records and in that one we will definitely talk about wider-scale application and potential geographical disparities. Here, it is more of a proof-of-concept paper. Overall, the paper could be improved greatly by adding more discussion of the methods, their results and their interpretation. At present the paper focuses a lot on the listing of the results of the different statistical analysis. This takes away from the potential interest of both the method and the results to the wider paleo-climate community. Author Response: We agree. However, as stated above, this is the first application of this method to a dataset of this kind and we want to focus here on the statistical and mathematical correctness and robustness of the method. In the subsequent paper we intend to talk more broadly about application for the broad community. One major concern is, that the manuscript is lacking a clear description of the data set that has been used, the way it was generated, and how this affects the analysis presented here. I know that this description is given in the original publication. Nevertheless, I think this is a vital point given that Lambert et al. (2012) state to keep the generation of the dust flux reconstruction in mind when interpreting its variance as it is affected by the assumptions and corrections that were involved. Author Response: We added a new paragraph (2.4) in the methods section that describes the dataset and it's features in relation with our analysis: "The dust flux data used in this study is based on a linear combination of insoluble particles, calcium, and non-sea-salt calcium concentrations (Lambert et al., 2012). Because missing data gaps in the three original datasets were linearly interpolated prior to the PCA, high frequency variability can sometimes be underestimated in short sections that feature a gap in one of the three original datasets. This occurs in about 25% of all dust flux data points, although half of those are concentrated in

the first 760 m of the core (0-43 kaBP), when an older less reliable dust measuring device was used. Below 760m these occurrences are evenly distributed and do not affect our analysis. Due to the sometimes slightly underestimated variability, the analysis shown here is a conservative estimate." One other major concern that I have is, that the interpretation of the, I think interesting, results unfortunately seems to be ad-hoc and not very thoroughly argued. From my reading of the previously rejected version of this paper and its reviews this point has only been improved marginally. To strengthen the manuscript and make it more suitable for Climate of the Past, I hope the authors extent the discussion of the results both in comparison with other studies and in terms of their paleo-climatological interpretation. The discussion and the results sections are completely lacking any information on the uncertainties of the obtained results. The large variability of the results between the different glacial/interglacial cycles indicates to me, that the results might not be very robust. One further observation that I made is that in many Figures the authors omit error or variability indications "for the sake of clarity" which I think is a poor choice. Additionally, the analysis is hinged upon a number of assumptions that are not justified in the text. Specifically, the slopes used for the breakpoint analysis and the range of time scales used for the fluctuation analysis. The influence of these choices on the results needs to be shown and clearly discussed. Author Response: We have added a new paragraph in the conclusion section about uncertainty estimates: "The results presented in this paper are largely empirical characterizations of a relatively less known source of climate data: dust fluxes. Dust flux statistics defy standard models: they require new analysis techniques and better physical models for their explanation. These reasons explain why our results may appear to be rough and approximate. Readers may nevertheless wonder why we did not provide standard uncertainty estimates. But meaningful uncertainties can only be made with respect to a theory and we have become used to theories that are deterministic, whose uncertainty is parametric, and that arises from measurement error. The present case is quite different: our basic theoretical framework is rather a stochastic one, it implicitly involves a stochastic "earth process" that produces an infinite number of statistically

identical planet earths of which we only have access to a single ensemble member. From this single realization, we neglected measurement errors and estimated various exponents that characterized the statistical variability over wide ranges of time scale, realizing that the exponents themselves are statistically variable from one realization to the next. In place of an uncertainty analysis, we therefore quantified the spread of the exponents (which themselves quantify variability). In the absence of a precise stochastic model we cannot do much better."

Specific comments: Abstract: P1 L17: The dataset has only a maximum resolution of 5 years. How can fluctuations on the one-year time scale be resolved. Please rephrase. Author Response: We changed this sentence to "The temporal resolution ranges from annual at the top of the core to 25 years at the bottom ,..." P1 L24-27: The logic of this sentence is not clear. Please rephrase. Author Response: Changed to "In other words, our results suggest that glacial maxima, interglacials, and glacial inceptions were characterized by relatively stable atmospheric conditions, but punctuated by frequent and severe droughts, whereas the mid-glacial climate was inherently more unstable." P1 L27f: Why do they suggest this? Author Response: We're putting in plain text what is stated in technical words in the previous sentence. Introduction: P2 L12-15: Please state that you refer to the temperature proxy time series, and also mention that this is a proxy, not a temperature in the strict sense. Author Response: Changed to "Fig. 1 shows this visually for the EPICA Dome C Antarctic ice core temperature proxy (5787 measurements in all);" P2 L23-27: Please consider explaining why the macro weather to climate transition timescale is important. Author Response: We added this: "The macroweather-climate transition scale marks a change of regime where the dominant high frequency processes associated with weather processes and reproduced by GCMs in control runs gives way to a new regime where the variability is dominated by either the responses to external forcings or to new, slow internal sources of variability." Method: P3 L15: ...the spectrum is the Fourier transform... Author Response: corrected P3 L24-26: It is unclear why due to scale invariance, the results from the dust fluxes can be transferred to the temperature proxies if they are affected

by different climatic mechanisms. Author Response: Scale invariance is a symmetry under time dilations. In a dynamical regime in which two different components such as temperature and dust are strongly coupled, each may have different scaling properties, but both should respect the scale symmetry including the transition scale at which the symmetry breaks down. It is only these aspects that can be "transferred". We have changed the text to reflect these ideas. P3 L26: The analysis in the "future publication" should more thoroughly be discussed here, especially as you present results from it, or alternatively only be mentioned in the outlook. In my experience these "future publications" unfortunately often do not manifest themselves. Author Response: The key analyses – the systematic comparison of temperature and dust as a function of time scale - have already been performed but this paper is already long enough. We prefer to discuss them in a separate paper. P4 L 24: extra comma between "compare" and "the" Author Response: corrected Results: The data set description should be in the Data + Methods section and more time should be spent describing the dataset used as pointed out above Author Response: We moved the data description to a new subchapter 2.4 and describe the data in detail as suggested. The new chapter 2.4 now reads "The dust flux data used in this study is based on a linear combination of insoluble particles, calcium, and non-sea-salt calcium concentrations (Lambert et al., 2012). Because no-data gaps in the three original datasets were linearly interpolated prior to the PCA, high frequency variability can sometimes be underestimated in short sections that feature a gap in one of the three original datasets. However, the amount of points that feature Unlike water isotopes that diffuse and lose their temporal resolution in the bottom section of an ice core at high pressures and densities, the relatively large dust particles diffuse much less and have been used to estimate the dust flux over every centimetre of the 3.2 km long EPICA core (298,203 measurements, (Lambert et al., 2012)). The temporal resolution of this series varies from 0.81 years to 11.1 yrs (the averages over the most recent and the most ancient 100 kyrs respectively). The worst temporal resolution of 25 years per centimeter occurs around 3050 m depth, with the result that at that resolution, there are virtually no missing data points in the whole

record (Fig. 1)." The last paragraph in this text was moved from sub-chapter 3.1. P6 L27: With the strong emphasis that is put on the number of datapoint and their sampling frequency throughout the manuscript, the numbers should add up. Generally, this information is not strictly necessary for the paper and could be safely removed. Especially in light of the fact that the dust flux reconstruction is a combination of a multitude of measurements and ice flow modelling. Author Response: The rest are data gaps due to cleaning of the raw data (contamination, missing ice, etc.). We changed the text to "298,203 valid measurements" P7 L1: Water isotopes cannot be assigned to one particular atmospheric variable either, even though they are often used to reconstruct Temperature. Please consider rephrasing Author Response: We changed that phrase to "Polar dust flux measurements cannot be assigned to one particular atmospheric variable, like temperature for the water isotopes." P7 L5ff: Please consider mentioning the recent publications by Markle et al (2018) and Schüpbach et al. (2018) that deal with this the relation of en-route washout and aerosol deposition on the ice sheets more quantitively. Author Response: Yes, maybe a distinction between short and large timescales is in order here. We changed these sentences to "At any given moment, the amount of dust deposited in East Antarctica will depend on the size and vegetation cover of the source region (mostly Patagonia for East Antarctic dust (Delmonte et al., 2008)), on the amount of dust available in the source region (can depend on the presence of glaciers), on the strength of the prevailing winds between South America and Antarctica, and the strength of the hydrological cycle (more precipitation will wash out more dust from the atmosphere (Lambert et al., 2008)). Over large scales it is thought that temperature-driven moisture condensation may be the major process driving low-frequency variability (Markle et al., 2018), although that may not globally be the case (Schüpbach et al., 2018)." P7 L19: There is no red line in the Figures. Author Response: Yes, we removed the word "red". P8 L7f: The unit of the spectral amplitude of the log-transformed fluxes is wrong. Author Response: Thank you, corrected. P8 L10f: Consider moving the comparison of the spectral densities with the results of the fluctuation analysis to after their introduction or to the discussion section. Author Response: Comparison moved to the discussion of Figure 5. P8 L13ff: This list of scaling exponents is completely irrelevant here and should be removed for clarity. Author Response: Sentence removed P8 L16f: It is unclear why this supports the use of dust as a proxy for atmospheric variability. Please clarify. Author Response: We argued that temperature and dust variability are of the same statistical type yet with significant differences. The former makes it likely that the dust signal is a real climate signal while the latter shows that it has different information. We eliminated the original sentence and replaced it with one similar to the above. P8 L 25-30: The results presented here are very hard to follow. Please consider reformulating. Author Response: We have modified the paragraph. P9 L1: Whether the Haar fluctuations of the dust flux have simple interpretations is not shown in the Figure but is rather a matter of taste and should be left to the reader to decide. Please reformulate or remove this statement. Author Response: Changed to "Haar fluctuations allow some direct interpretations..." P9 L6f: After spending a lot of time describing and interpreting the spectral analysis in the previous section the description here is rather short. As the fluctuation analysis is the main focus of this study the authors should spent more time describing their results and leading the uninitiated reader through them. Author Response: Yes, this is a good point, thanks. We replaced this first paragraph by three paragraphs giving a fuller explanation of the figure.

P9 L14: "positive definite" seems to be the wrong phase here, consider replacing with "always positive" or similar. Author Response: Corrected. P9 L18: Why is it important that the value is similar to those obtained from other ice cores. Please move all comparisons to other studies to the discussion clarify the interpretation and relevance. Author Response: We have removed that section in the revised manuscript. P9 L20: This statement is somewhat superfluous: If the dust fluxes are not log-normally distributed due to the occurrence of large spikes, their logarithm will not be normally distributed. Author Response: Yes. But due to the prevalence of log-transforming dust data in climate science, we would like to stress that statement. P9 L24: Please move the interpretation and the relevance for tipping point analysis to the discussion

and consider expanding on this point if you think it is an important application. Author Response: We have moved this sentence to the discussion. Discussion of this point would be a (actually several) paper in itself and is not central to this paper. But we feel that it is an important point to make for the tipping point community, many of which will read this paper. P9 L30f: Strictly, the statement that the scaling spectrum is the underlying behavior is an assumption and has not been shown in this study. Even though this a reasonable assumption, consider rephrasing. Author Response: Yes, we only show that the data are consistent with this hypothesis. No more can be done. We added a sentence to this effect. P10 L4f: Please clearly state that any of the stacking approaches assume that all the glacial cycles and their sub phases are realizations of the same underlying process. Author Response: We added "..., assuming that the major underlying processes were constant over the last 800,000 years" to that phrase. P10 L21ff: The start of this paragraph makes the reader expect spectra averaged over the different cycles. Please consider rephrasing and more extensively introducing the Figure. Author Response: We have modified this paragraph as suggested. P10 L29ff: The Figure indicates a slope of 0.35, not 0.25 as mentioned in the text. More importantly it is entirely unclear why the authors choose to set the slopes before and after the transition as constants and then only fit the transition time scale. The effect of the chosen values on the presented results needs to be clearly discussed and the uncertainty of the results are missing completely from the text. I strongly urge the authors to do a proper breakpoint and error analysis. Author Response: The value 0.25 was a typographic error, the value used was actually 0.35 as indicated in fig. 9 (not 10 as indicated in the text). The hypothesis here was that there were two regimes, each characterized by a different exponent each of which was estimated from the ensemble statistics. Given the hypothesis, the analysis only needed to estimate the scale at which the low frequency process exceeded the high frequency one. Therefore, we found the break point that minimized the RMS deviation from the bi-power law behaviour. We have added material explaining this better. P11 L5-9: As the other method is arguably only slightly more objective than the breakpoint inference by hand,

please consider removing this entire section. Author Response: We are dealing with a system with strongly nonclassical statistical characteristics: scaling and strongly non-Gaussian fluctuations. We do not have a precise stochastic model of the dynamics, the aim of this study is to yield a first, initial empirical characterization that could in the future provide the basis of a more precise model that would allow us to justify the estimation of uncertainty limits. P11 L20f: I suggest that the authors remove a couple of lines from the figure for clarity instead of the uncertainties or find a different way of visualizing the results. Author Response: Perhaps the problem is that the dashed lines were mistaken for uncertainty limits of the solid lines? We have added a phrase in the text to clarify this. There was also a reference two lines earlier to fig. 10 that should have been fig. 9. This correction may also help understand fig. 11. P11 L23: Please state why this exact time range was chosen. Author Response: As indicated in the text, the idea here was to estimate exponents from fixed ranges rather than ranges that varied depending on somewhat uncertain estimates of ïĄťc. The range of time scales was chosen so that in most phases, most of the range of time scales was in the climate regime (H>0), hence the lower limit of ïĄĎt>500 year. The upper limit of ïĄĎt < 3000 years was chosen because at longer scales, the statistics were less reliable: for phases 12500 years longer, there are only 4 disjoint intervals available and for larger ïĄĎt there are fewer. The range choice was a compromise that aimed at quantifying systematic phase to phase changes (the solid lines connecting the points) as well as the cycle to cycle dispersion of the exponents at each phase (the error bars). P12 L10: There is nothing black in Figure 12, please correct. Author Response: Corrected with additional information. P12 L18-21: Consider moving this sentence to the beginning of the paragraph to make it easier for the reader to follow the results. Author Response: Moved as suggested Discussion: P13 L7-9: You do not perform any significance analysis, so please reformulate this statement. Also please discuss why the lack of power around the obliquity cycle is surprising. Author Response: We replaced the precise word "significantly" by the vaguer term "barely" which is adequate for our purpose here. It is true that the absence of a 41kyr cycle is no longer surprising, we

have modified the sentence accordingly. P13 L28-32: Is there any supporting evidence for the glacier variability? Author Response: We added the reference Sugden et al., 2009 and Garcia et al., 2018 on South American glacial glacier variability. P14 L1: If A reflects the amplitude of the variance a change indicates only a change in the variability, not in their overall abundance, please reformulate. Author Response: Since dust is positive definite, a higher variability amplitude will produce a higher average. We added more explanation to this sentence: "Since the Argentinean continental shelf was still submerged at that moment and the outwash plains not yet fully extended, the higher dust emissions may have been due to a transformation in vegetation cover about 30 kyr after glacial inception, possibly accompanied by changes in glacial and periglacial processes in the Andes." P14 L14-17: Please discuss this statement in the context of recent proxy and model studies that indicate fast Southern Hemisphere circulation changes during DO events. (Markle et al. 2016, Buizert et al. 2018, Pedro et al. 2018). Author Response: We have expanded this paragraph, which now reads "The recuperation of vegetation cover would be more gradual, though, resulting in a saw-tooth shape of the dust spike that we do not observe in the data. Similarly, it has been suggested that rapid climate change in the Northern Hemisphere (e.g. Dansgaard/Oeschger events) would have synchronously changed the Southern Hemisphere atmospheric circulation and wind belts (Buizert et al., 2018; Markle et al., 2017). This could again have quickly changed the source or transport conditions, but would again have resulted in a saw-tooth shaped peak, either by steady regrowth of vegetation in the dust source areas, or as climate conditions in the north Atlantic gradually return to stadial (Pedro et al., 2018)." P14 L24: Only ice cores have the intrinsic property that they become less resolved with increasing depth and thus age. Sediment cores are not affected by this. Please reformulate. Author Response: In this sentence we only address the low (compared to the dust record used here) resolution of existing paleotemperature timeseries, not the change in resolution. We changed "temperature reconstructions" to "Pleistocene temperature reconstructions" to clarify. P14 L30: The paper does not show that the data neither over-samples nor smoothes.

[Figure]

Please either add this to the paper or remove this statement. Author Response: We deleted that last sentence and changed the previous one to "..., we therefore took advantage of the unique EPICA Dome C dust flux dataset with 1 cm resolution measuring 320,000 cm, whose worst time resolution over the whole core is 25 years." P15 L24: The reduction to the characterization of the different phase is a decision that the authors take, not an intrinsic property or the result of some analysis. Please reformulate this statement. Author Response: We changed this sentence to "We addressed the task of statistically characterizing the cycles by primarily characterizing the phases' variability exponents H, C1, qD and amplitude A." P15 L25: Missing dot between "A" and "We" Author Response: Corrected P15 L 26f: How is the variability of the dust flux at Dome C connected to the "activity" of the Patagonian ice sheet. Please extent and clarify this argument. Author Response: We removed the mention to the Patagonian ice sheet and changed this sentence to "However, the low amplitude of dust variability during glacial inceptions indicates that vegetation cover and dust production processes did not significantly change until ∼30 kyr after glacial inception." Figures: 6b: The units for the axis are wrong. Author Response: Yes, thanks, we also fixed the far-right expression for the number distribution. We have now indicated the units in the caption. 11: I suggest that the authors remove a couple of lines from the figure for clarity instead of the uncertainties or find a different way of visualizing the results. Author Response: We responded to this suggestion above.

Please also note the supplement to this comment:
https://www.clim-past-discuss.net/cp-2018-171/cp-2018-171-AC3-supplement.pdf

---

## Author Response (AR2)

**Editor**

Dear Dr Lovejoy,

I read the comments of the experts after the second run of review. Although most of the critical point raised have been clarified in your review version, I agree that there is still a number of concern to be dealt with before that the MS can be published in Climate of the Past. I think that the question raised by one of the reviewer and about the dust flux measurement is quite relevant and I strongly appreciate if you can improve the MS in this sense.

Overall the MS reads well, it is now much clearer, and your arguments are more easily followed and especially the description of the results and the discussion.

I look forward to see the re-revised version of the MS.

Best wishes,
Carlo Barbante

**Michel Crucifix**

The authors have addressed all comments point-by-point. Results are significant and the study should be published. I am still left with a couple of comments, all are mostly editorial comments, or at least, can be addressed with adequate edition.

p.2: "indicating that the new long frequency processes become dominant". Not clear what is the "new" long-frequency process (in what sense is it 'new' and does it 'become' dominant". Perhaps be more explicit about what understood as of anthropogenic origin?

*Authors:*

*The word "new was perhaps unfortunate, what was meant was "different" from the higher frequency processes, we now added: "that were too weak to be important at higher frequencies" to make this even more clear. It is now stated "In the last century, anthropogenically forced temperature changes (mostly from greenhouse gases) dominate the natural (internal, macroweather) variability at scales longer than about about 10- 20 years"*

p.4 ("respect the scale symmetry") and p.11 ("Scaling is a statistical symmetry"). Turbulence theory (as, among others, given in the Lovevoy / Schertzer book, ch. 2) tell us how symmetry causes scaling, but the article here misses a proper definition of what is a "scale symmetry", or in what sense "scaling is a statistical symmetry". Isn't it more accurate to state that "(statistical) scaling is a consequence of (physical) symmetry"?

*Authors:*

*We were trying to make the point about scaling symmetries being broken on each realization, only holding exactly on an infinite ensemble. We have now modified the opening sentences of section 3.4 to read:*

*"Scaling is a statistical symmetry, a consequence of a time and space scaling symmetry of the underlying dynamics. Being statistical means that on average the statistics at small, medium and large scales are the same in some way (more precisely, it holds over a statistical ensemble)."*

p.8 : "that may not globally be the case" : be more accurate or explicit : "although this may not be the case                                                                                                    everywhere".

*Authors: We improved the sentence accordingly.*

p. 11, l. 3: "in dimensional or nondimensional time that statistical" -> "in dimensional or nondimensional time, statistical"

*Authors: fixed*

p. 10, l. 12: remove second occurrence of "clearly"

*Authors: fixed*

p. 16, l. 32: "thanks to the scaling, very few processes are Gaussian". Deterministic theory is very at ease with explaining non-Gaussian fluctuations without resorting to scaling arguments. The Lorenz 63 system maps Gaussian-distributed initial conditions onto a non-Gaussian distribution of final states.

*Authors: Yes, we added : " nonlinear dynamics"*

p. 20, ll. 17- 24 : This author response to my earlier comment left me a bit unsatisfied. It is commonplace in statistics to provide estimators and estimator variances for stochastic processes, a problem which can generally be addressed with either a frequentist or a Bayesian framework. Perhaps the most straightforward and classical example is the estimation of the mean of a random process, but other trivial and less trivial examples include the problem of estimating the coefficients of an auto-regressive processes, spectrum estimation, parameter estimation of non-linear stochastic dynamical systems.... So if I understand correctly the author's problem is the lack of a reasonable 'generic stochastic process' model, which they could derive a distribution from. This seems rather different than the objection they raise, which they claim is related to the problem of having a single draw from a stochastic process.

*Authors: Yes, of course both points are pertinent, and we do discuss your point.*

*We added the sentence ". Unfortunately, we do not yet have a good stochastic process model from which we can infer sampling errors. "*

*We have also added earlier in the text some references to some of the limited uncertainty results that are available for various exponents.*

**Reviewer 3**

General remarks and major comments:

Overall, the Authors have greatly improved the presented manuscript in the second version. It is now much clearer, and the arguments of the Authors are more easily followed and especially the description of the results and their discussion has won a lot. However, before the publication of the paper there are a few more issues that should be addressed. Especially the second one is still major in my view.

The Authors always refer to "dust flux measurements" multiple times throughout the manuscript. Even though this might be a small detail, I think it is misleading as the dust flux is not directly measurable but a derived quantity that incorporates both measurements of dust concentrations as well as (model) inferred accumulation rates. Even worse, the dust flux data used here is the result of a combination of a range of different dust and dust proxy measurements. I suggest the authors reformulate for example to dust flux data, record or reconstruction.

*Authors: Corrected throughout the manuscript*

Thank you for adding the description and discussion of the data set that was employed in section 2.4 of the paper. One major thing, however, is still missing: The dust flux reconstruction uses 55 cm resolution accumulation rate reconstructions that are based on the empirical conversion of deuterium isotopic ratios of the water to accumulation rates. For analysis of variability beyond the resolution of 55 cm this poses a serious issue: Any variability beyond the resolution of the accumulation rate reconstruction is purely a result of the variability of the dust concentration in the ice. In turn this means that if the variability of the dust flux is interpreted one interprets different records at different time scales, i.e. fluxes at scales >55 cm and concentrations at scales <55 cm. Especially due to the fact that the time scales represented by these depth resolutions changes with depth in the core the effect on the fluctuation analysis is not entirely obvious and likely not straight forward.
The authors should thoroughly test the effect of this on their results and clearly discuss the effect in the paper.

*Authors: There are two issues here, one being the contribution of the accumulation rate to the variability, and the other is the different effect at various depths due to ice layer thinning. In the following figure we have plotted in blue the Haar fluctuation for both Ca and nssCa concentrations (multiplied with a constant mean accumulation rate for the units to match). In black we have the dust fluxes (thick line is the complete record, dashed and dotted lines are specific cycles as indicated). We can see that the effect of the lower resolution accumulation rate is a slightly less steep gradient in the dust flux than the calcium concentrations, which indeed adds to the uncertainty of our exponents, although we now run into the question of representativeness of calcium as a dust proxy. We have added the phrase "We note, however, that the dust flux used here is a construct of concentrations at 1cm resolution and accumulation rates at 55 cm resolution that were linearly interpolated to match the dust concentration resolution." in chapter 2.4, and the sentences "We should also mention that the use of fluxes (produce of 1cm concentrations*

*and 55cm accumulation rate) introduces an additional source of uncertainty due to the different time ranges contained in these sections at various depths. However, we prefer using the fluxes because they are more directly representative of climatic changes than concentrations." In chapter 3.3 to mention this. Since our main results are not impacted, we don't think an in-depth analysis is warranted at this point.*

[Figure]

*The interpretation of our results at various depths due to the varying resolution of the record is already implicitly included in Figure 9, although it was not discussed. We have now added some cycle numbers in the right-hand panels (cycles) and added the sentence ".  Here, the same colours and numbers correspond to the cycle number, shown are only cycles 1, 4, 8 are indicated to avoid clutter." The point here is that if there was an effect with depth, the lines for the cycles would be ordered, from first to last or vice-versa. Since the cycles are quite randomly distributed, we can be sure there is no effect due to the depth of the cycle within the ice core. We added the sentence "We could also note that since the different cycles had quite similar statistics (the right hand column), that this implies that there is no bias in the flux estimates with depth of the core." in the main text discussing this figure.*

The discussion of the uncertainties related to the analysis that was added to the conclusions is better

placed in the discussion, in my opinion. Furthermore, I feel that the "we live in a single realization" argument is a little bit of an easy answer to the concerns raised in the first round of reviews. I will leave it up to the editor, whether a more thorough uncertainty analysis using i.e. simulations needs to be added or if the presented argument is sufficient.

*Authors: See our answer to a similar comment by Michel Crucifix*

Minor comments:

P2L25ff: In the added explanation of the macroweather-climate transition scale, the authors elude twice to "new long frequency processes" and "new, slow internal sources of variability" without going into much more detail in the remainder of the paper. I suggest to either at least clearly hypothesize and discuss the nature of these "new" processes and sources of variability or remove the "new" from each of the sentences.

*Authors: See our response to same remark by Michel Crucifix*

P14L26ff: Where do you show the change in correlation, is the reference missing or is this planned in a future publication?

*Authors: Indeed, that is a figure from our next publication. Since it's still in prep we have deleted this sentence from the text.*

P16L31ff: The whole paragraph about DO-events is repeated from above, I assume this is an editing error.

*Authors: Fixed*

[revised manuscript text omitted]